# SeBA: Semi-supervised few-shot learning via Separated-at-Birth Alignment for tabular data

## Abstract

Learning from scarce labeled data with a larger pool of unlabeled samples, known as semi-supervised few-shot learning (SS-FSL), remains critical for applications involving tabular data in domains like medicine, finance, and science. The existing SS-FSL methods often rely on self-supervised learning (SSL) frameworks developed for vision or language, which assume the availability of a natural form of data augmentations. For tabular data, defining meaningful augmentations is non-trivial and can easily distort semantics, limiting the effectiveness of conventional SSL. In this work, we rethink SSL for tabular data and propose Separated-at-Birth Alignment (SeBA), a joint-embedding framework for SS-FSL that eliminates the dependence on augmentations. Our core idea is to separate the data into two independent, but complementary views and align the representations of one view to mirror the nearest-neighbor correspondence of the data in the second view. A type-aware separation scheme ensures robust handling of mixed categorical and numerical attributes, while a lightweight architecture with ensemble aggregation improves generalization and reduces sensitivity to misselection of model parameters. An experimental study conducted in various benchmark datasets demonstrates that SeBA often achieves state-of-the-art performance in tabular SS-FSL, opening a new avenue for SSL paradigm in the domain of tabular data.

## 1 Introduction

Learning with a limited amount of labeled data remains a fundamental challenge in machine learning and data analysis. Although collecting additional annotations is costly, access to raw unlabeled data is often inexpensive. This imbalance motivates the practical setting of semi-supervised few-shot learning (SS-FSL), where classification must be performed with scarce labeled data and a large pool of unlabeled samples (see Figure 1a). Applications in disease diagnosis (Shailaja et al., 2018), credit risk prediction (Clements et al., 2020), and cognitive sciences (Grabowska et al., 2025) highlight the need for approaches tailored to tabular data. Despite its importance, tabular SS-FSL has been underexplored compared to FSL methods for computer vision (CV) and natural language processing (NLP) (Finn et al., 2017; Snell et al., 2017; Sendera et al., 2023; Przewięźlikowski et al., 2022; Brown et al., 2020; Min et al., 2022).

The tabular modality poses unique challenges for typical SS-FSL methods, which rely on pretraining with unlabeled data followed by fine-tuning on a few labeled examples (see Figure 1). State-of-the-art pretraining approaches often use self-supervised learning (SSL), which encourages models to produce similar representations for semantically related *positive pairs* while avoiding collapse to trivial solutions (Wang & Isola, 2020). Such pairs are usually created by sampling multiple augmentations of the same data point. In CV, augmentations are straightforward: image transformations such as cropping, rotation, or color jittering yield valid semantically consistent samples. However, for tabular data, there is no natural way to define proper augmentations. Poorly chosen transformations, such as zero masking, Gaussian noise, or

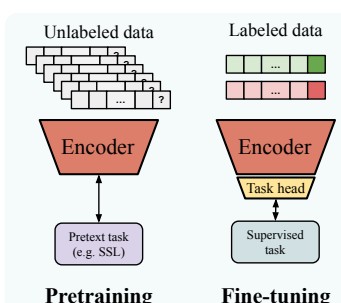

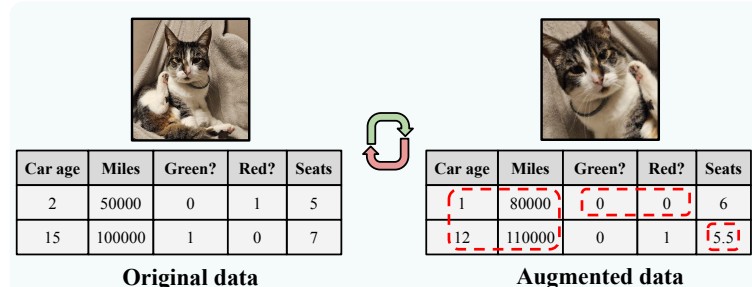

(a) Semi-supervised Few-shot Learning (SS-FSL) setup. The model is pretrained on a large pool of unlabeled data and next fine-tuned on a few labeled examples.

(b) While semantic-preserving augmentations are straightforward to define for modalities such as images, they must be designed much more carefully for tabular data. Improperly designed augmentations can generate samples from outside the data manifold (decreasing car age, but increasing mileage), obfuscate the categorical values (neither option marked as true), or assign incorrect values (number of car seats must be an integer).

Figure 1: Typical Semi-supervised Few-shot Learning (SS-FSL) approaches **(a)** pretrain their representations on large pools of unlabeled data, usually via Self-supervised Learning (SSL). In the case of tabular data, the state-of-the-art augmentation-based SSL approaches cannot be directly applied, due to challenges with defining proper augmentations **(b)**. In this work, we introduce Separated-at-Birth Alignment, which removes the need for augmentations and improves the tabular representations for SS-FSL (see Figure 2).

sampling features from marginal distribution, can distort semantics or even generate out-of-distribution samples (see Figure 1b), ultimately undermining the effectiveness of SSL. Consequently, recent work has turned to alternative pretraining strategies for tabular representation learning – such as cluster detection with prototypical networks (Nam et al., 2023), or diffusion-based methods (Liu et al., 2024) – largely sideline SSL.

In this paper, we rethink SSL for tabular data and show that, with carefully designed positive pairs, it yields significantly stronger tabular representations than previously assumed. Instead of aligning the representations of positive pairs created via augmentations, we introduce Separated-at-Birth Alignment (SeBA), illustrated in Figure 2. SeBA projects data into two complementary subspaces: feature and target views. The model is then pretrained by identifying nearest-neighbor correspondences in the target view, using only the information encoded in the feature view. This replaces the problematic reliance on augmentations with a nearest-neighbor graph. To properly handle mixed data types, we employ a type-aware separation scheme that accounts for both categorical and numerical features, ensuring that the resulting projections remain semantically meaningful.

SeBA requires far less dataset-specific knowledge than hand-crafting augmentations, making it practical and easy to apply. Moreover, the model pretrained by SeBA is lightweight and thus less prone to overfitting for small datasets. Finally, the applied ensemble strategy minimizes the need for the selection of model's parameters, which is crucial in the FSL scenario. The experimental results clearly show that SeBA generalizes effectively in a wide variety of tabular datasets, achieving impressive results in few-shot classification.

**Our contributions can be summarized as follows:**

- We introduce Separated-at-Birth Alignment (SeBA), a novel self-supervised pretraining approach for tabular Few-shot Learning that replaces augmentation-based positive pair construction with a nearest-neighbor graph, removing the need for hand-crafted augmentations.

- We instantiate SeBA as a lightweight model with an ensembling strategy that prevents overfitting for small datasets and minimizes the need for the selection of model parameters in the case of few labeled examples.

- We provide an in-depth empirical analysis that confirms the validity of our approach.

- We experimentally verify that SeBA generalizes across a wide variety of tabular datasets, achieving strong performance in few-shot classification.

## 2 RELATED WORK

**Self-supervised learning for tabular data**   One of the first approaches to self-supervised learning for tabular data, VIME (Yoon et al., 2020), creates a novel pretext task of estimating mask vectors from corrupted tabular data in addition to the reconstruction pretext task for self-supervised learning. (Bahri et al., 2021) propose SCARF, a contrastive learning method in which different views of a sample are obtained by corrupting a random subset of features. In both methods, data corruption is implemented as sampling from empirical distribution on masked features. (Ucar et al., 2021) propose SubTab that divides the input features into multiple subsets to perform a pretext task close to mask reconstruction and contrastive learning simultaneously. (Sui et al., 2023) propose an augmentation-free method that simultaneously reconstructs multiple randomly generated data projection functions to generate a data representation. T-JEPA (Thimonier et al., 2025) is also an augmentation-free method that is trained by predicting the latent representation of a subset of features from the latent representation of a different subset within the same sample. PTaRL has investigated prototype-based representation learning for tabular data (Ye et al., 2024), which first constructs a prototype-based projection space (P-Space) and then learns the disentangled representation around the global data prototypes. Most of these methods are applied as pretraining techniques, where several deep classifiers are trained on the obtained representation, which is not well suited to SS-FSL problem considered here.

**Few-shot learning for tabular data**   Some recent research claims that SSL methods cannot create representations for tabular data that can be fine-tuned to target tasks with a limited number of labels (Nam et al., 2023; Liu et al., 2024). As a remedy, (Nam et al., 2023) propose STUNT, a meta-learning method that pretrains a ProtoNet on self-generated tasks using only unlabeled data. The authors show that given only a few labeled samples per class, STUNT outperforms other methods. (Liu et al., 2024) propose D2R2 to extract representations by training a conditional diffusion model and aligning the distances from various random projection spaces. Although D2R2 reports the SOTA results, they are obtained in a transductive setting, in which the model is tuned according to the information coming from the test set. We are the first who show that inferior performance of SSL methods in a few-shot setting is not inherent to tabular data. The proposed construction of SeBA allows us to construct a representation that is state-of-the-art in tabular few-shot learning.

**Transfer learning and other pretraining techniques**   Other works, such as XTab (Zhu et al., 2023), TransTab (Wang & Sun, 2022), or UniTabE (Yang et al., 2023), propose a cross-table pretraining approach for building representation. A seminal work of (Hollmann et al., 2023) introducing TabPFN demonstrated how to pretrain a transformer-based architecture on synthetically generated data. Once the model is pretrained, it can be applied in a zero-shot setting to new data or further fine-tuned on target task (Breejen et al., 2024). In contrast to the transformer-based methods mentioned above, SeBA uses a lightweight MLP and a single linear layer as the task-specific head, which can be fine-tuned effectively using only a few labeled samples.

## 3 METHOD

**Problem statement and scope.**   We consider a semi-supervised few-shot learning problem (SS-FSL), illustrated in Figure 1a. In the training stage, we have access to a small labeled support set $\mathcal{S} = \{(s_i, y_i)\}_{i=1}^{N_s}$ and a large portion of unlabeled data $\mathcal{D} = \{(u_i)\}_{i=1}^{N_u}$, where $s_i, u_i \in \mathbb{R}^D$ are feature vectors, $y_i \in \{1, 2, \ldots, C\}$ denotes the class label and $C$ is the number of classes. We assume that $N_u \gg N_s$. Furthermore, in the $N$-way $K$-shot setting, each of the $N$ classes in the support set is represented by exactly $K$ labeled samples. Our goal is to train a model $h_\theta : \mathbb{R}^D \to \{1, 2, \ldots, C\}$ for the classification task defined by the support data. In a typical inductive setting considered here, we evaluate the model on the query set $\mathcal{Q} = \{q_i\}_{i=1}^{N_q}$ of previously unseen data.

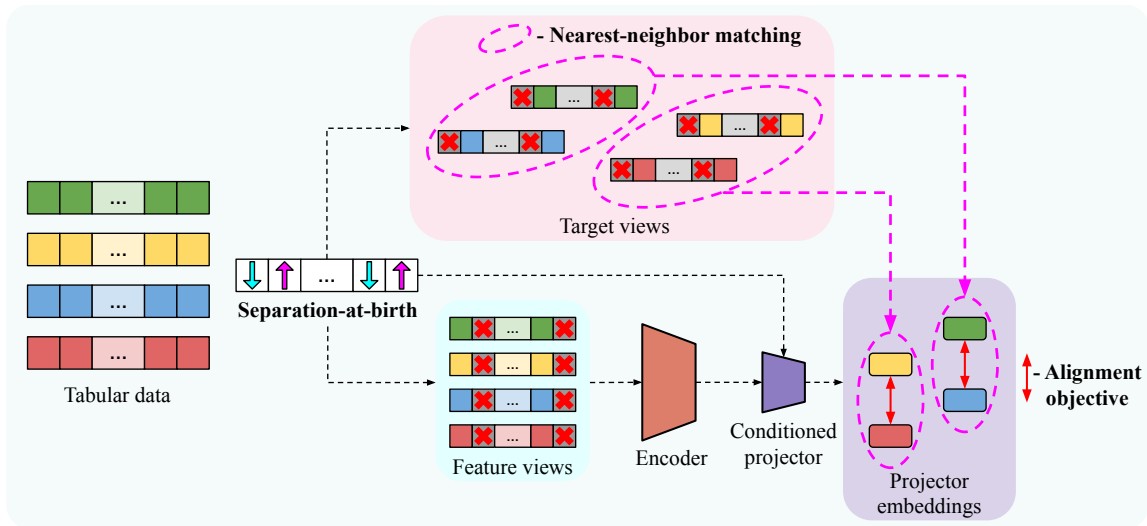

Figure 2: Representation learning via Separated-at-Birth Alignment (SeBA). In each minibatch, we separate the columns of tabular data "at-birth" into two complementary and independent subsets, which define target and feature views. Instead of augmentation, semantically-related positive pairs for a pretraining contrastive task are defined using the nearest neighbor relation in their target view (upper side). The encoder is trained to create the representation of the feature view, which aligns the positive pairs defined in the target view (bottom side). To allow the encoder to create general data representation, SeBA uses a conditioned projector to build a task-specific representation for every separation mask.

The typical SS-FSL training divides the training of $h_\theta$ into two phases: **(i)** unsupervised pretraining of the encoder $f : \mathbb{R}^D \to \mathbb{R}^E$, and **(ii)** supervised training of a classifier $c : \mathbb{R}^E \to \mathbb{R}^C$ on $\mathcal{S}$ on top of the representation created by $f$, where $E$ is the embedding shape of the encoder (Nam et al., 2023; Liu et al., 2024). Therefore, the final model is given by $h_\theta = c \circ f$.

In this work, we follow the standard protocol for training the few-shot classifier and focus primarily on the pretraining phase, where we introduce a novel approach called Separated-at-Birth Alignment (SeBA), described below.

### 3.1 SEPARATED-AT-BIRTH ALIGNMENT (SEBA)

**Overview.** The design of SeBA follows Self-supervised Joint-Embedding Architectures (JEAs), which learn through aligning semantically-related positive pairs of data in the representation space (Chen et al., 2020; He et al., 2020) and pushing away the unrelated ones. Unlike conventional JEAs, SeBA does not rely on sampling multiple data augmentations for the construction of positive pairs, which is problematic for tabular data. Instead, in every minibatch, SeBA separates the tabular records "at birth" into two random complementary views, which we denote as feature and target views. The model is trained to align the data representations of feature views according to the similarity graph induced by the target views, therefore, learning semantically meaningful correspondences without relying on augmentations. We outline the schema of SeBA in Figure 2 and describe its components in detail below.

**Separating the data at-birth.** Let $\mathcal{B}$ be a minibatch of (unlabeled) data points, and let $m \in \{0, 1\}^D$ be a binary mask vector sampled once per batch that defines features included in each view. The proportion of 1-s

in $m$ is controlled by the hyperparameter named *separation ratio*. For every $x \in \mathcal{B}$, we create a feature view:

$$x_f = x \odot (1 - m) \tag{1}$$

and a target view:

$$x_t = x \odot m, \tag{2}$$

where $\odot$ is an element-wise multiplication, and $x_t, x_f \in \mathbb{R}^D$. As such, the target and feature views are complementary and independent.

**Target similarity graph.** We use target views to define the positive data pairs with respect to the sampled mask $m$. For each sample $x$ in the batch, we identify its nearest neighbor in terms of the target views:

$$x' = \arg \min_{a \in \mathcal{B} \setminus \{x\}} d(x_t, a_t). \tag{3}$$

In other words, $m$ defines positive pairs $(x, x')$ based on the nearest-neighbor graph defined in the target view.

In Appendix B, we present a theoretical justification that our approach produces positive pairs, which are meaningful for solving a target task. Given Gaussian classes, we consider a probability of the mismatch event that a sample and its nearest neighbor calculated in a target view come from different classes. We derive that the expected value of this probability decays exponentially as the dimensionality of target view and the separation between Gaussians increase. In consequence, we are allowed to perform nearest neighbor search in target view to produce meaningful positive pairs without the risk that the nearest neighbor will have a different class.

**Alignment objective.** Finally, we train the encoder to align the feature-view representations to match the nearest-neighbor relation defined in the target view. For this purpose, we first construct the encoder representations of the feature views:

$$h = f(x_f); h' = f(x'_f) \tag{4}$$

Observe that the feature views $x_f$ do not contain unequivocal information about the data separation scheme $m$ and, as such, it may not be able to solve the alignment task on their own. To address this problem, SeBA incorporates a conditioned projector $\pi : \mathbb{R}^E \times \mathbb{R}^D \to \mathbb{R}^P$, where $P$ is the embedding shape of $\pi$ (Przewięźlikowski et al., 2024; Bordes et al., 2023). The projector combines the general feature view representation of the encoder and information about how the data were separated (i.e. the mask vector $m$). The projector transforms the encoder representation into the task-specific latent space:

$$z = \pi(h, m); z' = \pi(h', m), \tag{5}$$

in which we optimize the alignment objective. The objective takes form of the InfoNCE loss, which pulls together the positive representation pairs, and pushes away the unrelated ones (Oord et al., 2018):

$$\mathcal{L}(x) = -\log \frac{\exp(d(z, z'))}{\Sigma_{a \in \mathcal{B}, a \neq x} \exp\left(d\Big(z, \pi\big(f(a_f), m\big)\Big)\right)} \tag{6}$$

Because SeBA trains on numerous separation schemes (multiple mask vectors), the encoder adapts repeatedly to different target matching objectives. This exposure yields robust representations that generalize well to downstream tasks.

### 3.2 PRETRAINING DETAILS

The proposed pretraining benefits from several design choices, discussed in the following and ablated in Section 4.4.

**Type-aware feature preprocessing and separation.**   We normalize the numerical columns by removing the mean and scaling to unit variance, and encode the categorical data using one-hot vectors. A straightforward separation applied to that representation could result in splitting a single categorical variable between the two views, even though all of its coordinates represent the same variable. The proper way to take into account the specific nature of the categorical variable, is to define the separation mask $m$ in the original representation (before the one-hot encoding), which prevents splitting a single category into two views.

**Missing data encoding.**   When separating the data into feature and target views, the question arises of what to replace the separated features with. Although this issue could be eliminated in a transformer-based architecture, it must be addressed with our chosen lightweight MLP encoder which is more suitable for small datasets (Thimonier et al., 2025). In contrast to previous works, which commonly sampled masked features from empirical distribution (Nam et al., 2023), or trained a dedicated missing data imputation module (Smieja et al., 2018), we find that zero imputation works best in our case.

**Separation ratio.**   To allow the encoder to create a general representation that suits multiple tasks, we could iterate over all possible mask vectors when separating the data into views. However, feeding the network with tabular masked records with variable mask size could hurt its performance (Yi et al., 2020). Consequently, we constrain the selection of the mask by fixing the number of its non-zero elements $|m| = \sum_{i=1}^{D} m_i$ to a constant value $T$.

**Ensembling.**   Since it could be difficult to choose a single value of $T$ that fits all datasets, we decided to generate multiple representations, each trained using a different value of $T$. In other words, for every $T \in \{T_1, \ldots, T_k\}$, we train individual SeBA models that are next fine-tuned for a specific target task. In inference, we aggregate their predictions using the ensemble method. In this way, we eliminate the need of defining optimal masking ratio $T$ in general or tuning this parameter for each data set, which could be impossible due to the limited number of labeled examples.

## 4 EXPERIMENTS

### 4.1 EXPERIMENTAL SETUP

We follow the benchmark proposed by (Nam et al., 2023) and then developed by (Liu et al., 2024) verifying the performance of the models in a few-shot learning scenario.

**Datasets preparation.**   In addition to the 8 datasets used in previous SS-FSL benchmarks (Nam et al., 2023), we select 4 more datasets from the OpenML-CC18 benchmark (Asuncion et al., 2007; Bischl et al., 2017), see Appendix A.1 for details. Following Nam et al. (2023), all datasets are randomly divided into train and test sets in a ratio of 5:1. The training data are treated unlabeled and are used for model pretraining. In addition, 10% of the training data is used for validation. Once the model is pretrained, it is fine-tuned on the support set and evaluated on the query set. The support and query sets are randomly sampled from the test set. In the $N$-shot $K$-way setting, the support set contains $N$ examples of each of $K$ classes. We consider 1-, 5-, and 10-shot settings.

**Setup of SeBA.**   We pretrain the encoder and projector of SeBA for 10 000 epochs, using the early stopping. We stop training when the value of the objective function, measured on the validation set, stops decreasing for 100 epochs. Following (Nam et al., 2023), the encoder is a 2-layer MLP with a hidden dimension of 1024, and the projector is also a 2-layer network with the same hidden dimension and an embedding dimension of 256, a common choice in contrastive learning (Chen et al., 2021). In the fine-tuning stage, we freeze the encoder and train a classification head at the top of the encoder representation using the support set. For 5-

Table 1: Evaluation in terms of 1-shot classification accuracy.

| Method | CMC | DIA | DNA | INC | KAR | OPT | PIX | SEM | GES | MAR | SAT | TEX |
|--------|-----|-----|-----|-----|-----|-----|-----|-----|-----|-----|-----|-----|
| CatBoost | 36.03 | 56.74 | 39.15 | 57.55 | 53.24 | 58.30 | 54.74 | 43.21 | 23.90 | 53.42 | 58.21 | 47.23 |
| kNN | 35.39 | 58.50 | 42.20 | 51.45 | 54.61 | 65.60 | 60.79 | 44.35 | 24.57 | 55.31 | 60.67 | 61.07 |
| TabPFN | 35.37 | 53.35 | 41.83 | 49.30 | 46.02 | 55.74 | 23.79 | 28.01 | 25.95 | 58.25 | 55.29 | – |
| SubTab | 36.23 | 58.22 | 46.98 | 62.45 | 50.22 | 62.01 | 60.34 | 39.99 | **27.19** | 59.18 | 56.12 | 59.81 |
| VIME | 35.90 | 58.99 | 51.23 | 61.82 | 59.81 | 69.26 | 63.28 | 46.99 | 25.58 | **60.10** | 61.54 | 50.79 |
| Scarf | 35.39 | 55.64 | 57.86 | 57.94 | 60.96 | 63.31 | 63.93 | 29.39 | 18.81 | 49.72 | 55.28 | 30.50 |
| UMTRA | 35.46 | 57.64 | 25.13 | 57.23 | 49.05 | 49.87 | 34.26 | 26.33 | – | – | – | – |
| SES | 34.59 | 59.97 | 39.56 | 56.39 | 49.19 | 56.30 | 49.19 | 33.73 | – | – | – | – |
| T-JEPA | 34.28 | 50.44 | 36.14 | 48.97 | 30.22 | 39.84 | 39.89 | 25.47 | 25.13 | 50.72 | 55.18 | 56.77 |
| Pseudo-label | 34.97 | 57.03 | 44.26 | 60.52 | 49.44 | 61.50 | 56.12 | 41.42 | 23.43 | 52.85 | 60.32 | 62.92 |
| Mean Teacher | 35.58 | 58.05 | 46.58 | 60.63 | 54.57 | 66.10 | 61.02 | 43.56 | 23.40 | 47.67 | 52.71 | 54.66 |
| SAINT | 35.41 | 59.65 | 36.88 | 54.69 | 40.13 | 52.92 | 24.25 | 32.52 | 22.56 | 47.55 | 44.45 | 56.71 |
| CACTUs | 36.10 | 58.92 | 65.93 | 64.02 | 65.59 | 71.98 | 67.61 | 48.96 | – | – | – | – |
| STUNT | 37.10 | 61.08 | 66.20 | 63.52 | 71.20 | 76.94 | 79.05 | 55.91 | 27.04 | 53.88 | 63.12 | 58.69 |
| D2R2-c | **40.81** | 60.10 | 61.29 | **72.85** | 61.45 | 77.41 | 61.45 | 34.26 | 26.58 | 51.70 | 60.92 | 61.78 |
| SeBA (our) | 36.76 | **61.14** | **66.79** | 62.89 | **76.40** | **78.94** | **83.06** | **61.11** | 27.07 | 58.43 | **65.70** | **70.94** |

and 10-shot, we use linear probing, while for 1-shot setting, we assign query samples to the closest class prototypes based on the support set.

Experiments on 1- and 5-shot classification are repeated with 100 different random seeds, while in the case of 10-shot learning, we use 20 seeds. A higher number of seeds reduces the randomness related to model initialization and dataset splits. For each train-test split, we sample the support and query sets 100 times and average the accuracy metrics over all splits and all selections of the support/query sets.

## 4.2 FEW-SHOT CLASSIFICATION

We evaluate SeBA in terms of its performance in downstream few-shot learning tasks. We compare our method with the state-of-the-art SS-FSL methods, STUNT (Nam et al., 2023), and D2R2[1] (Liu et al., 2024), which we run on exactly the same splits as SeBA. We also report the results of 13 other baselines from (Nam et al., 2023). They represent the best supervised, self-supervised and meta-learning approaches (see Appendix A.2 for more details).

We present the results of the evaluation in 1-, 5-, and 10-shot classification in Tables 1 to 3, respectively. The efficacy of SeBA increases consistently with the number of support examples, as opposed to approaches like D2R2-c, which exhibit significant variance in quality on datasets like SEM. SeBA achieves the best accuracy in 23 out of 36 instances and the second-best in 9 out of the remaining 13, which further confirms its practicality and applicability to a wide range of datasets. Furthermore, in Table 10 we present the standard deviations of results of 3 best-performing methods (STUNT, D2R2-c, and SeBA) and find that the results of SeBA vary less than that of D2R2-c and STUNT, highlighting the greater stability of our method. We summarize the average performance of each method in Figure 5, from which it is evident that SeBA is generally the best-performing approach.

---

[1]We use an inductive variant of D2R2, i.e. D2R2-c, which uses mean support embeddings as the classifier (Liu et al., 2024). The default D2R2 uses an instance-wise iterative prototype scheme, additionally using query data for class prototype estimation. This is not consistent with the inductive setting, where queries are unseen during classifier training.

Table 2: Evaluation in terms of 5-shot classification accuracy.

| Method | CMC | DIA | DNA | INC | KAR | OPT | PIX | SEM | GES | MAR | SAT | TEX |
|---|---|---|---|---|---|---|---|---|---|---|---|---|
| CatBoost | 39.89 | 64.51 | 60.20 | 67.99 | 77.94 | 83.07 | 83.38 | 68.69 | 32.01 | 64.78 | 73.13 | 76.06 |
| kNN | 37.65 | 65.61 | 61.16 | 62.19 | 80.08 | 84.16 | 84.75 | 68.33 | 26.86 | 59.47 | 71.25 | 71.14 |
| TabPFN | 38.31 | 64.06 | 52.72 | 64.11 | 76.59 | 81.68 | 62.41 | 56.20 | 31.51 | 64.49 | 77.77 | – |
| SubTab | 39.81 | 68.26 | 62.49 | 72.14 | 70.88 | 83.27 | 80.41 | 59.87 | 29.38 | **67.47** | 73.67 | 71.11 |
| VIME | 39.83 | 67.64 | 71.29 | 72.19 | 19.42 | 83.21 | 85.24 | 68.45 | 29.44 | 64.50 | 72.30 | 59.99 |
| Scarf | 37.75 | 68.66 | 62.75 | 66.09 | 69.96 | 85.67 | 81.32 | 35.20 | 30.12 | 48.97 | 64.61 | 32.42 |
| UMTRA | 38.05 | 64.41 | 25.08 | 65.78 | 67.28 | 73.29 | 51.32 | 35.90 | – | – | – | – |
| SES | 39.04 | 66.61 | 52.25 | 68.27 | 74.80 | 78.46 | 74.80 | 52.74 | – | – | – | – |
| T-JEPA | 37.25 | 64.06 | 43.04 | 62.13 | 60.07 | 71.93 | 73.66 | 50.56 | 30.84 | 55.61 | 77.34 | 84.65 |
| Pseudo-label | 37.49 | 64.46 | 60.06 | 66.26 | 78.60 | 83.71 | 82.94 | 67.49 | 31.59 | 59.17 | 78.39 | **92.10** |
| Mean Teacher | 37.73 | 65.45 | 61.47 | 67.05 | 81.08 | 86.66 | 85.24 | 69.67 | 31.78 | 58.23 | 78.52 | 85.47 |
| SAINT | 35.41 | 66.94 | 46.34 | 60.01 | 71.07 | 79.53 | 40.35 | 62.42 | **33.29** | 54.87 | 75.87 | 82.56 |
| CACTUs | 38.81 | 66.79 | 81.52 | 72.03 | 82.20 | 85.92 | 85.25 | 65.00 | – | – | – | – |
| STUNT | 40.40 | **69.88** | 79.18 | 72.69 | 85.45 | 88.42 | 89.08 | 71.54 | 32.19 | 58.62 | 74.25 | 68.57 |
| D2R2-c | **43.39** | 68.69 | **81.39** | 73.34 | 79.49 | 87.12 | 82.22 | 60.16 | 30.26 | 56.24 | 70.66 | 71.82 |
| SeBA (our) | 42.85 | 69.54 | 79.86 | 71.28 | **87.59** | **90.11** | **91.88** | **79.41** | 32.07 | 65.22 | **78.66** | 87.51 |

Table 3: Evaluation in terms of 10-shot classification accuracy.

| Method | CMC | DIA | DNA | INC | KAR | OPT | PIX | SEM | GES | MAR | SAT | TEX |
|---|---|---|---|---|---|---|---|---|---|---|---|---|
| STUNT | 42.01 | 72.82 | 80.96 | 74.08 | 86.95 | 89.91 | 89.98 | 74.74 | 34.30 | 61.08 | 75.58 | 71.10 |
| D2R2-c | 37.86 | 72.02 | 81.72 | **75.34** | 84.81 | 89.27 | 73.70 | 36.30 | 31.22 | 59.80 | 71.96 | 73.43 |
| SeBA (our) | **46.30** | **73.61** | **83.59** | 72.68 | **90.88** | **92.62** | **93.88** | **84.11** | **34.60** | **69.96** | **81.17** | **90.18** |

## 4.3 Alignment of the SeBA pretraining objective with recognition task

In this section, we evaluate the validity of the proposed Separated-at-Birth Alignment as an unsupervised pretraining objective. For this purpose, we analyze its stability and the semantic relationship of the positive pairs created by SeBA. For each dataset, we generate 100 random separations into feature and target views with a separation ratio of 0.2. Next, we identify the nearest neighbors of the samples in terms of target views (see eq. (3)).

To measure how the SeBA objective aligns with the downstream classification task, we measure the proportion of pairs in which the nearest neighbors share the same class as the original instance, see Figure 3a, and the proportion of same-class samples within the increasing number of nearest neighbors, see Figure 3b, for the SEM dataset. A detailed inspection (the results for the remaining datasets are shown in Figure 6 and Figure 7) reveals that the vast majority of nearest-neighbor pairs share the same class, which indicates that the pretraining objective learns features useful for a downstream task, and that selecting the immediate nearest neighbor as the target yields the best chance of pairing same-class samples.

We also verify the stability of SeBA. To this end, we count the number of unique samples that are matched to a given instance as nearest neighbors. It is evident from Figure 3c that the average number of unique neighbors for the SEM dataset reaches up to 5% of the entire data, showing low noise in the pretraining objective,

see Figure 8. Finally, we verify that with separation ratios in our chosen range (0.1-0.5), the proportions of semantically meaningful sampled nearest-neighbor pairs remain high (see Figure 3d and Figure 9).

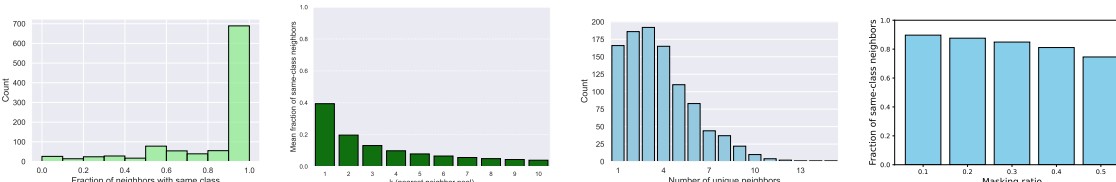

(a) Same-class neighbor ratios.

(b) Fraction of same-class samples among $k$ nearest neighbors.

(c) Unique neighbor number distribution.

(d) Fraction of same-class neighbors sampled under different separation ratios.

Figure 3: Analysis of neighbor stability under masked perturbations for Semeion dataset: (a) high fraction of neighbors sharing the same class label as the original instance confirms high consistency between pretext and downstream tasks, (b) the fraction of same-class samples among the k nearest neighbors decreases as k increases, (c) low number of unique neighbors for each sample indicates high stability of SeBA, and (d) lower separation ratio yields a larger chance of sampling same-class nearest neighbor pairs.

## 4.4 ABLATION STUDY

In this section, we ablate the design choices of SeBA: data preprocessing, separation ratio, and the choice of the multi-shot classifier. The model variants are evaluated in terms of 5-shot classification accuracy with 5 random seeds. We detail the model variants and report the results in Tables 6 to 8, and summarize them in Figure 4.

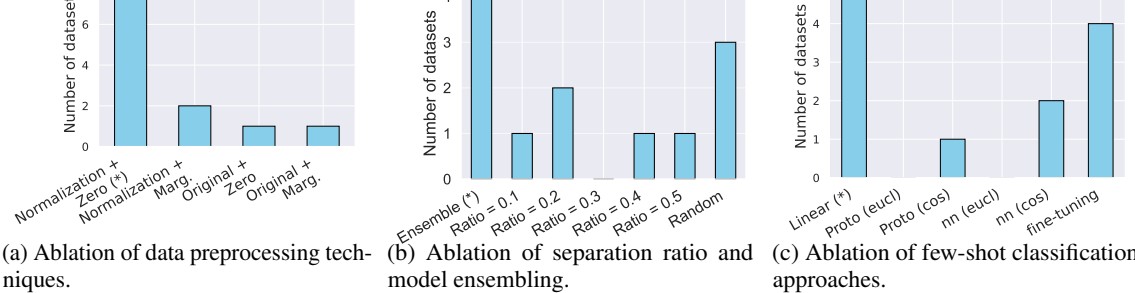

(a) Ablation of data preprocessing techniques.

(b) Ablation of separation ratio and model ensembling.

(c) Ablation of few-shot classification approaches.

Figure 4: Ablation of the design aspects of SeBA ((**\***) denotes the default setting of SeBA). In the barplots, we report the number of datasets in which a given variant of SeBA performs best.

**Data preprocessing (Figure 4a / Table 6).** We ablate the usefulness of data normalization and two variants of missing data imputation: zero filling and sampling column values from marginal distribution. In most cases, the combination of data normalization and zero imputation yields representations of the highest quality.

**Separation ratio and model ensembling (Figure 4b / Table 7).** We ablate the choice of target / feature separation ratio and compare it with an ensemble of encoders trained with all ratios, as well as randomly sampling the mask ratio during training. Although certain ratios and random sampling yield the best results for several datasets, we find that ensembles of encoders perform most reliably.

**Learning the few-shot classifier (Figure 4c / Table 8).** We compare several choices of learning the classifier on top of the pretrained representation from the support data in the multi-shot setting. Our analysis shows that linear probing is the simplest and most effective approach.

## 5 CONCLUSION

In this paper, we introduce Separated-at-Birth Alignment (SeBA), a novel Semi-supervised Few-Shot Learning framework designed for tabular data. SeBA uses the powerful Joint-Embedding Architecture (JEA) paradigm to pretrain its representations, while avoiding the problematic need to for manual data augmentation design – the issue that has prevented the use of JEAs for tabular data in the past. Instead, our core idea is to separate the data "at birth" into two independent, complementary subspaces and align the representations of one subspace to mirror the nearest-neighbor correspondence of the data in the second subspace. We demonstrate that this pretraining task indeed captures the semantic correspondence in a wide variety of tabular datasets. SeBA achieves impressive performance in few-shot learning on various tabular datasets, confirming its effectiveness. Our findings open new avenues for further investigations into tabular representation learning and are a useful foundation for data-constrained applications.

**Ethics statement.** This paper presents work whose goal is to advance the field of Machine Learning. There are many potential societal consequences of our work, none of which we feel must be highlighted here.

**Reproducibility statement.** We have described all the details and hyperparameters of the proposed approach. We include our codebase as supplementary material and will publish it along with the paper.

**LLM statement.** The authors used LLM tools to polish the writing.

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

## A   EXPERIMENT DETAILS

### A.1   DATASETS

The details of the datasets are presented in Table 4.

### A.2   BASELINES

Along with SeBA, we report the performance of fourteen methods taken from (Liu et al., 2024), which represent three types of baselines:

1. **Supervised.** CatBoost (Prokhorenkova et al., 2018) is considered a shallow SOTA approach to tabular data; k-NN (Peterson, 2009) works well for the few-shot case; TabPFN (Hollmann et al., 2023) represents a transformer-based zero-shot technique, which can be applied to small datasets.

2. **Self-supervised.** VIME (Yoon et al., 2020), SubTab (Ucar et al., 2021) and SCARF (Bahri et al., 2021), T-JEPA (Thimonier et al., 2025), and SAINT (Somepalli et al., 2021) represent the typical SSL approaches for tabular data. The representations acquired from those models are used to conduct Center Prototype Classification or linear probing.

3. **Few-shot / meta-learning.** Although UMTRA (Sun et al., 2019), SES (Ye et al., 2022) and CACTUs (Hsu et al., 2018) are designed for image data, they were repurposed for tabular data modality by Nam et al. (2023). Since the source code for both of these implementations is not publicly available, we report the results originally reported in Nam et al. (2023).

Table 4: Overview of the datasets used in the experiments, including the number of instances, proportion of numerical and categorical features, and the number of classes.

| Dataset code | Dataset | # Instances | # Features (num., cate.) | # Classes |
|---|---|---|---|---|
| CMC | cmc | 1473 | 9 (2,7) | 3 |
| DIA | diabetes | 768 | 8 (8,0) | 2 |
| DNA | dna | 3186 | 180 (0,180) | 3 |
| INC | income | 48842 | 14 (6,8) | 2 |
| KAR | karhunen | 2000 | 64 (64,0) | 10 |
| OPT | optdigits | 5620 | 64 (64,0) | 10 |
| PIX | pixel | 2000 | 240 (0,240) | 2 |
| SEM | semeion | 1593 | 256 (256,0) | 10 |
| GES | GesturePhaseSegmentationProcessed | 9873 | 32 (32,0) | 5 |
| MAR | bank-marketing | 45211 | 16 (5,11) | 2 |
| SAT | satimage | 6430 | 36 (36,0) | 6 |
| TEX | texture | 5500 | 40 (40,0) | 11 |

4. **Semi-supervised learning**. Mean Teacher Tarvainen & Valpola (2017) is a method which uses the consistency loss between the teacher output and student output. Pseudo-label Lee et al. (2013) predicts the labels of unlabeled data and then uses them as learning signal.

STUNT (Nam et al., 2023) and D2R2 (Liu et al., 2024) were evaluated using authors' repositories with hyperparameters selection procedures implemented there. For D2R2, we run the variant denoted by D2R2-c, which uses mean support embeddings as the classifier. The default D2R2 uses an instance-wise iterative prototype scheme, additionally using query data for class prototype estimation. This is not consistent with the inductive setting, where queries are unseen during classifier training.

### A.3 Hyperparameters

We report the values of the hyperparameters used by SeBA in Table 5.

### A.4 Implementation details

We implement SeBA in PyTorch Paszke et al. (2019). We include the codebase as supplementary material and will publish it along with the paper. All of the experiments described in the paper were run on a single NVidia-V100 GPU.

## B Theoretical analysis

### B.1 Summary of the results

We present a theoretical analysis, which confirms that, on average, nearest neighbors in target views give samples of the same class.

For transparency, let us consider a simplified model of data on $\mathbb{R}^D$ representing two Gaussian classes:

$$X \mid Y = c \sim \mathcal{N}(\nu_c, I), \qquad X \mid Y = c' \sim \mathcal{N}(\nu_{c'}, I).$$

Working with arbitrary covariances (but identical for both classes) is possible but requires data whitening, which further complicates the derivation. Therefore, we restrict our attention to the isotropic case. Let us

Table 5: SeBA hyperparameters

| Hyperparameter | Value |
|---|---|
| Pretraining | |
| Epochs | 10.000 |
| Learning rate | 0.001 |
| Optimizer | Adam (Kingma & Ba, 2014) |
| Batch size | 1024 |
| Early stopping patience | 100 |
| Encoder depth | 2 |
| Encoder hidden size | 1024 |
| Encoder output size | 256 |
| Projector depth | 2 |
| Projector hidden size | 1024 |
| Projector output size | 256 |
| Few-shot classification | |
| Epochs | 10.000 |
| Learning rate | 0.001 |
| Optimizer | Adam (Kingma & Ba, 2014) |

denote the class-mean difference $\Delta = (\nu_{c'} - \nu_c)$. We can calculate the squared separation between Gaussian means as:

$$\delta^2 = \Delta^T \Delta = \sum_{i=1}^{D} \Delta_i^2.$$

We are interested in the probability of the *mismatch* – the event that the nearest neighbor of a sample $x$ has a different class than $x$. Observe that the probability of mismatch decreases as the separation becomes larger.

Thus, we are interested in preserving separation in target views, where the nearest neighbors are calculated. If we consider a $n$-dimensional target view defined by a subset $S \subset \{1, \ldots, D\}$ with $|S| = n$, then the squared separation in this view equals:

$$\delta_S^2 = \sum_{i \in S} \Delta_i^2.$$

As can be seen, we cannot guaranty that every target view provides a well-separated class-mean difference. However, we can show that with small probability we select a target view, where the probability of mismatch is large.

Our analysis consists of two parts:

1. We bound the probability of mismatch in the target view $S$ from the above by the exponential term dependent on the separation $\delta_S^2$ in this view.

2. Taking the expectation, we show that on average, the probability of mismatch in the target view decays exponentially in the separation $\delta^2$ in the original data space $\mathbb{R}^D$.

The last fact is crucial because it guaranties that the separation is preserved on average if only initial classes are also separated enough. In consequence, we are allowed to perform nearest neighbor search in target view to produce meaningful positive pairs without the risk that the nearest neighbor will have a different class.

## B.2 Bound on the mismatch probability

**Lemma B.1.** *Let $X$, $Y$ be independent draws from $\mathcal{N}(\mu, I_d)$ (the same class), and let $Z$ be an independent draw from $\mathcal{N}(\mu + \Delta, I_d)$ (a different class), where $\Delta \in \mathbb{R}^d$ denotes the mean offset. Define the mismatch event*

$$\mathcal{E} \;=\; \{\, \|Z - X\|^2 \leq \|Y - X\|^2 \,\},$$

*i.e., the sample of the other-class $Z$ is no further than the sample of the same-class $Y$. There exists a global constant $C > 0$ such that*

$$\Pr(\mathcal{E}) \leq 2 \exp(-C\|\Delta\|^2)$$

*Proof.* Denote

$$X = \mu + \varepsilon_X, \qquad Y = \mu + \varepsilon_Y, \qquad Z = \mu + \Delta + \varepsilon_Z,$$

with $\varepsilon_X, \varepsilon_Y, \varepsilon_Z \overset{\text{i.i.d.}}{\sim} \mathcal{N}(0, I_d)$. Set

$$U := \varepsilon_Z - \varepsilon_X, \qquad V := \varepsilon_Y - \varepsilon_X.$$

Then $U$ and $V$ are independent vectors of $\mathcal{N}(0, 2I_d)$ and

$$D := \|Z - X\|^2 - \|Y - X\|^2 \;=\; \|\Delta + U\|^2 - \|V\|^2$$
$$= \|\Delta\|^2 + 2\Delta^\top U + (\|U\|^2 - \|V\|^2).$$

The mismatch event $\mathcal{E}$ equals $\{D \leq 0\}$, that is,

$$2\Delta^\top U + (\|U\|^2 - \|V\|^2) \leq -\|\Delta\|^2. \tag{1}$$

If the sum of two independent random terms is $\leq -\|\Delta\|^2$, then at least one of them must be $\leq -\frac{1}{2}\|\Delta\|^2$. Hence, by the union bound,

$$\Pr(D \leq 0) \leq \Pr\!\Big(2\Delta^\top U \leq -\tfrac{1}{2}\|\Delta\|^2\Big) + \Pr\!\Big(\|U\|^2 - \|V\|^2 \leq -\tfrac{1}{2}\|\Delta\|^2\Big). \tag{2}$$

We bound each term on the right-hand side.

**(i) Linear term.** The scalar random variable $G := 2\Delta^\top U$ is Gaussian with mean 0 and variance

$$\mathrm{Var}(G) \;=\; 4\mathrm{Var}(\Delta^\top U) = 4 \cdot \Delta^\top(2I_d)\Delta = 8\|\Delta\|^2.$$

Therefore, for $a = \frac{1}{2}\|\Delta\|^2$, the Gaussian tail bound gives

$$\Pr\!\Big(2\Delta^\top U \leq -a\Big) \leq \exp\!\Big(-\frac{a^2}{2\mathrm{Var}(G)}\Big) = \exp\!\Big(-\frac{(\frac{1}{2}\|\Delta\|^2)^2}{16\|\Delta\|^2}\Big) = \exp\!\Big(-\frac{\|\Delta\|^2}{64}\Big). \tag{3}$$

**(ii) Quadratic-difference term.** Note that $\|U\|^2$ and $\|V\|^2$ are independent and each has the distribution of 2 times a $\chi_d^2$ random variable, hence $\mathbb{E}\|U\|^2 = \mathbb{E}\|V\|^2 = 2d$. We bound the probability that their difference is $\leq -\frac{1}{2}\|\Delta\|^2$ by controlling the deviations of each chi-square from its mean. For any $t > 0$,

$$\Pr\!\Big(\|U\|^2 - \|V\|^2 \leq -t\Big) \leq \Pr\!\Big(\|U\|^2 \leq 2d - \tfrac{t}{2}\Big) + \Pr\!\Big(\|V\|^2 \geq 2d + \tfrac{t}{2}\Big).$$

Standard chi-square concentration (e.g. $\Pr(\chi_d^2 \leq d(1 - \epsilon)) \leq \exp(-d\epsilon^2/4)$ for $0 < \epsilon < 1$, and $\Pr(\chi_d^2 \geq d(1 + \epsilon)) \leq \exp(-d\epsilon^2/4)$ for $\epsilon > 0$) implies that, for $t = \frac{1}{2}\|\Delta\|^2$, both probabilities are bounded by $\exp(-\frac{\|\Delta\|^4}{256d})$ and, in consequence,

$$\Pr\!\Big(\|U\|^2 - \|V\|^2 \leq -\tfrac{1}{2}\|\Delta\|^2\Big) \leq 2\exp\!\Big(-\frac{\|\Delta\|^4}{256d})\Big). \tag{4}$$

Combining (2),(3),(4) yields

$$\Pr(D \le 0) \le \exp\Big( - \frac{\|\Delta\|^2}{64} \Big) + 2\exp\Big( - \frac{\|\Delta\|^4}{256d} \Big).$$

Finally, we may consider two cases: (i) $\|\Delta\|^2 \ge d$, and (ii) $\|\Delta\|^2 < d$. In both cases, we can find a common factor $C > 0$ such that

$$\Pr(D \le 0) \le 2\exp(-C\|\Delta\|^2)$$

which completes the proof. □

### B.3 Expected mismatch probability

Let $S \subset \mathbb{R}^D$ define the $n$ dimensional target view. By $P_S(X)$ we denote a coordinate-wise projection of a sample $X \in \mathbb{R}^D$ onto a target view $S$. Given two samples $X, Y \in \mathbb{R}^D$ from a class $c$, a other sample $Z \in \mathbb{R}^D$ from class $c' \ne c$, we define the probability of mismatch in a view $S$ as

$$p_{\text{mismatch}}(S) := \Pr\big( \|P_S(Y) - P_S(X)\|^2 \le \|P_S(Z) - P_S(X)\|^2 \big).$$

The following theorem shows that the expected mismatch probability in the target view decays exponentially in the preserved fraction $n/D$ of the full separation $\delta^2$ and it does not depend on the separation in the individual target views.

**Theorem B.1.** *Let us consider data on $\mathbb{R}^D$ representing two Gaussian classes:*

$$X \mid Y = c \sim \mathcal{N}(\nu_c, I), \qquad X \mid Y = c' \sim \mathcal{N}(\nu_{c'}, I)$$

*with the squared separation term $\delta^2 = (\nu_{c'} - \nu_c)^T(\nu_{c'} - \nu_c)$. Let $S$ be chosen uniformly at random among all $\binom{D}{n}$ $n$ dimensional target views. Then the expected probability of mismatch can be bounded as*

$$\mathbb{E}_S[p_{\text{mismatch}}(S)] \le 2\exp\Big( - C\frac{n\delta^2}{D} \Big),$$

*for some constant $C > 0$.*

*Proof.* The projected squared separation equals $\delta_S^2 = \sum_{i \in S} a_i$ and Lemma B.1 implies that

$$p_{\text{mismatch}}(S) \le 2\exp\Big( - C\delta_S^2 \Big),$$

for some constant $C > 0$.

Taking the expectation over $S$ and using Jensen's inequality ($e^{-x}$ is convex downward, so $\mathbb{E}[e^{-X}] \le e^{-\mathbb{E}[X]}$), we get:

$$\mathbb{E}_S[p_{\text{mismatch}}(S)] \le 2\exp\Big( - C\mathbb{E}_S[\delta_S^2] \Big).$$

Finally, $\mathbb{E}_S[\delta_S^2] = \frac{n}{D}\delta^2$ by the symmetry of uniform sampling without replacement, which gives:

$$\mathbb{E}_S[p_{\text{mismatch}}(S)] \le 2\exp\Big( - C\frac{n\delta^2}{D} \Big).$$

□

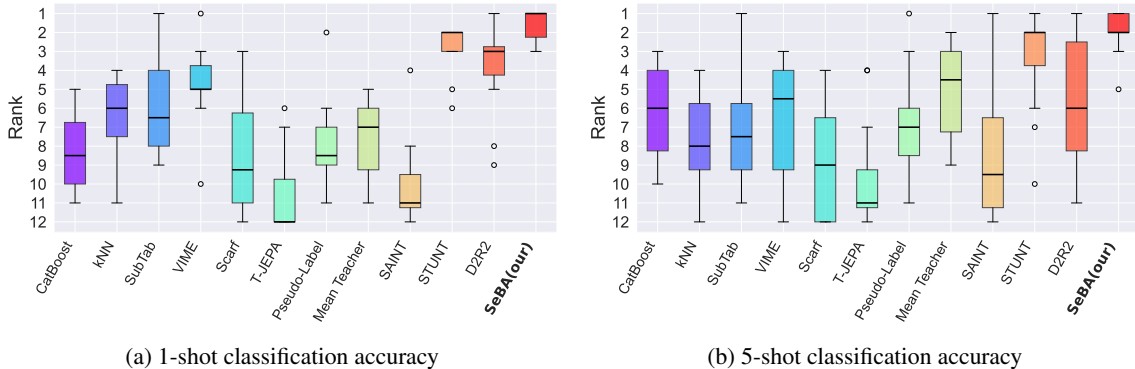

(a) 1-shot classification accuracy        (b) 5-shot classification accuracy

Figure 5: Box-plots of 1-shot (left) and 5-shot (right) classification ranks of benchmarked approaches. SeBA is the most consistently high-ranking method.

## C ADDITIONAL EXPERIMENTAL RESULTS

### C.1 FEW-SHOT LEARNING PERFORMANCE SUMMARY

We rank the performance of models in terms of 1- and 5-shot classification from Table 1 and Table 2 on 8 common datasets and report the rank distributions in Figure 5. It is evident that SeBA gives significantly better rank than competitive methods.

### C.2 DETAILED ABLATION STUDY RESULTS

In Tables 6 to 8, we include the details of variants and results of ablations described in Section 4.4. Furthermore, in Table 9 we compare the default variant of SeBA trained with a variant which uses an FT-Transformer architecture (Gorishniy et al., 2023) as an encoder. We find the MLP encoder architecture to achieve the better results.

Table 6: Ablation of normalizing the numerical columns in tabular data (**Norm.**) and of the type of data imputation in separated columns (**Imput.**), where we compare zero-imputation (Zero), and sampling from the column's marginal distribution (Marg.).

| Norm. | Imput. | CMC | DIA | DNA | INC | KAR | OPT | PIX | SEM | GES | MAR | SAT | TEX |
|---|---|---|---|---|---|---|---|---|---|---|---|---|---|
| True | Zero (*) | **42.85** | 69.54 | **79.86** | **71.28** | 87.59 | **90.11** | **91.88** | **79.41** | 32.07 | **65.22** | 78.66 | **87.51** |
|  | Marg. | 41.71 | **69.78** | 68.69 | 67.89 | 86.36 | 89.18 | 89.67 | 77.32 | **32.70** | 53.19 | 78.64 | 84.83 |
| False | Zero | 40.50 | 53.49 | 70.82 | 47.01 | **91.00** | 87.91 | 90.54 | 77.28 | 31.06 | 60.16 | 78.67 | 68.43 |
|  | Marg. | 37.95 | 54.97 | 69.71 | 46.44 | 89.09 | 89.41 | 89.29 | 77.85 | 30.87 | 59.40 | **78.69** | 79.04 |

### C.3 PRETRAINING AND FEW-SHOT ALIGMENT

We present detailed results for all datasets of the stability analysis performed in Section 4.3. In Figure 6, we report the proportion of immediate neighbors sharing the same class label across different separations, in Figure 7 we measure the average fraction of neighbors sharing the same class label as a given data sample,

Table 7: Ablation of the separation ratios between the target and feature data views, compared with the ensemble of encoders trained with different ratios and randomly sampled mask ratios.

| Mode | CMC | DIA | DNA | INC | KAR | OPT | PIX | SEM | GES | MAR | SAT | TEX |
|------|-----|-----|-----|-----|-----|-----|-----|-----|-----|-----|-----|-----|
| Ensemble (*) | 42.85 | 69.54 | **79.86** | **71.28** | 87.59 | 90.11 | **91.88** | 79.41 | 32.07 | 65.22 | 78.66 | 87.51 |
| Ratio = 0.1 | 41.18 | 68.20 | 69.07 | 69.73 | 84.21 | 85.84 | 89.37 | 72.02 | **32.21** | 64.98 | 78.53 | 85.08 |
| Ratio = 0.2 | 41.56 | **69.73** | 73.64 | 68.14 | 86.36 | 88.83 | 91.13 | 76.04 | 31.88 | 62.33 | 78.66 | **88.16** |
| Ratio = 0.3 | 42.71 | 68.19 | 77.01 | 68.40 | 87.53 | 90.29 | 91.24 | 77.33 | 31.80 | 63.58 | 78.21 | 87.98 |
| Ratio = 0.4 | 41.86 | 68.41 | 73.33 | 70.11 | 87.43 | **90.69** | 91.79 | 78.69 | 31.40 | 64.28 | 77.82 | 87.08 |
| Ratio = 0.5 | 41.16 | 68.26 | 72.69 | 70.74 | **88.11** | 90.51 | 89.83 | 78.94 | 31.35 | 59.66 | 77.87 | 85.37 |
| Random | **43.57** | 60.83 | 67.19 | 65.26 | 86.37 | 89.91 | 88.91 | 74.98 | 32.02 | **68.76** | **79.40** | 85.75 |

Table 8: Ablation of different ways of forming the many-shot classifier. We compare linear probing (Linear), using support data to form prototypes and assigning queries based on Euclidean od cosine distance (Proto eucl/cos), matching individual support representations as nearest neighbors based on Euclidean od cosine distance (nn eucl /cos), and fine-tuning the whole encoder along with the classifier (fine-tuning).

| Mode | CMC | DIA | DNA | INC | KAR | OPT | PIX | SEM | GES | MAR | SAT | TEX |
|------|-----|-----|-----|-----|-----|-----|-----|-----|-----|-----|-----|-----|
| Linear (*) | **42.85** | 69.54 | **79.86** | 71.28 | 87.59 | 90.11 | 91.88 | 79.41 | **32.07** | 65.22 | **78.66** | **87.51** |
| Proto (eucl) | 39.78 | 67.27 | 75.85 | 70.45 | 86.06 | 86.76 | 92.33 | 75.16 | 28.70 | 66.68 | 73.46 | 76.66 |
| Proto (cos) | 39.35 | 67.66 | 78.49 | 70.30 | 87.36 | 89.61 | **92.71** | 75.50 | 31.10 | 65.45 | 77.29 | 79.97 |
| nn (eucl) | 40.31 | 66.57 | 72.68 | 69.22 | 86.26 | 89.74 | 92.37 | 76.41 | 28.14 | 64.59 | 74.97 | 84.76 |
| nn (cos) | 40.83 | 67.34 | 74.53 | 70.14 | **87.74** | **91.09** | 92.55 | 77.74 | 30.41 | 63.57 | 77.78 | 86.28 |
| fine-tuning | 42.52 | **70.31** | 76.30 | **71.63** | 85.27 | 89.09 | 91.71 | **79.48** | 31.27 | **66.91** | 78.15 | 86.17 |

when inspecting exactly $k$ nearest neighbors, while Figure 8 shows the numbers of unique samples matched as nearest neighbors.

All metrics follow similar-shaped distributions for the majority of datasets, indicating high stability of the SeBA pretraining objective. The exceptions include the CMC and GES datasets, in which the performance of SeBA is relatively lower, especially in 1- and 5-shot classification tasks. This indicates that while SeBA is generally a good pretext task for learning tabular representation, there is still room for improvement in future work.

Table 9: Comparison of SeBA with MLP and FT-Transformer (Gorishniy et al., 2023) encoders in terms of 5-shot accuracy. The MLP architecture yields consistently stronger representations.

| Encoder | CMC | DIA | DNA | INC | KAR | OPT | PIX | SEM | GES | MAR | SAT | TEX |
|---|---|---|---|---|---|---|---|---|---|---|---|---|
| FT-Transformer | 39.02 | 64.69 | 73.17 | 69.51 | 84.14 | **91.47** | 86.63 | 78.59 | 30.63 | 60.22 | 78.02 | 79.70 |
| MLP (*) | **42.85** | **69.54** | **79.86** | **71.28** | **87.59** | 90.11 | **91.88** | **79.41** | **32.07** | **65.22** | **78.66** | **87.51** |

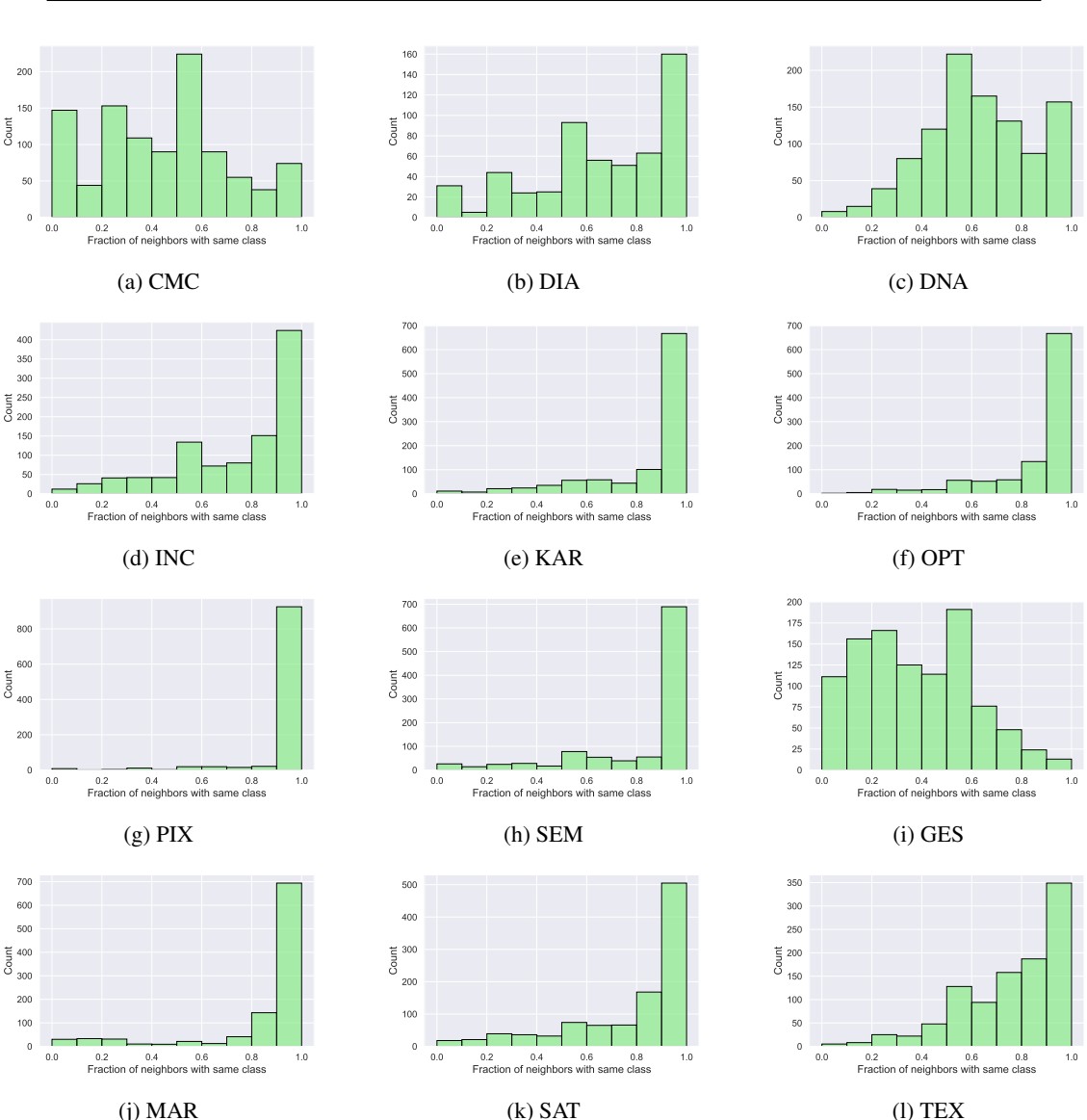

(a) CMC       (b) DIA       (c) DNA

(d) INC       (e) KAR       (f) OPT

(g) PIX       (h) SEM       (i) GES

(j) MAR       (k) SAT       (l) TEX

Figure 6: The fraction of neighbors sharing the same class label as the original instance. High values indicate high alignment between pretext and downstream tasks.

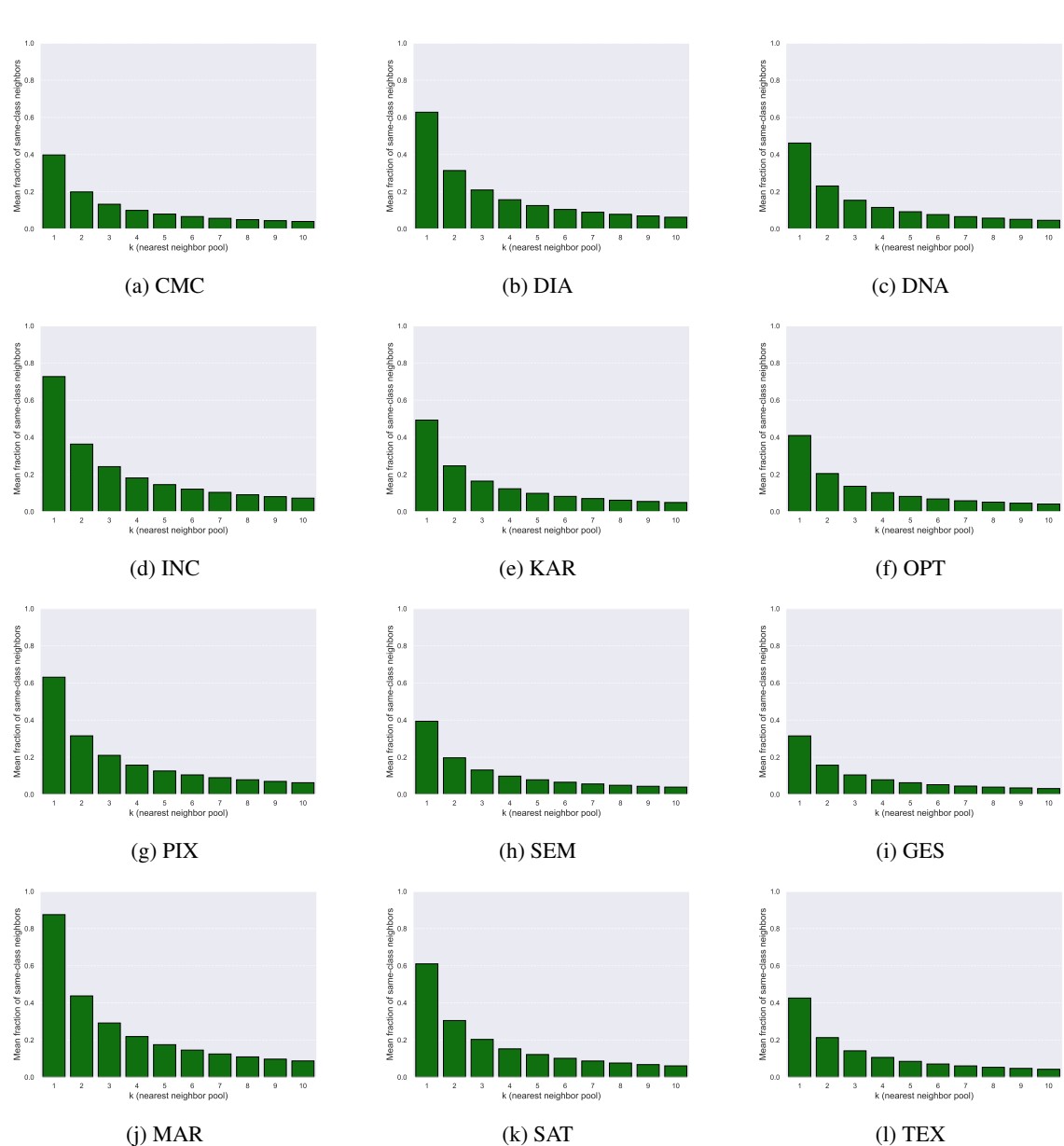

Figure 7: Average fraction of nearest neighbors sharing the same class label as a given data sample, when inspecting $k$ nearest neighbors.

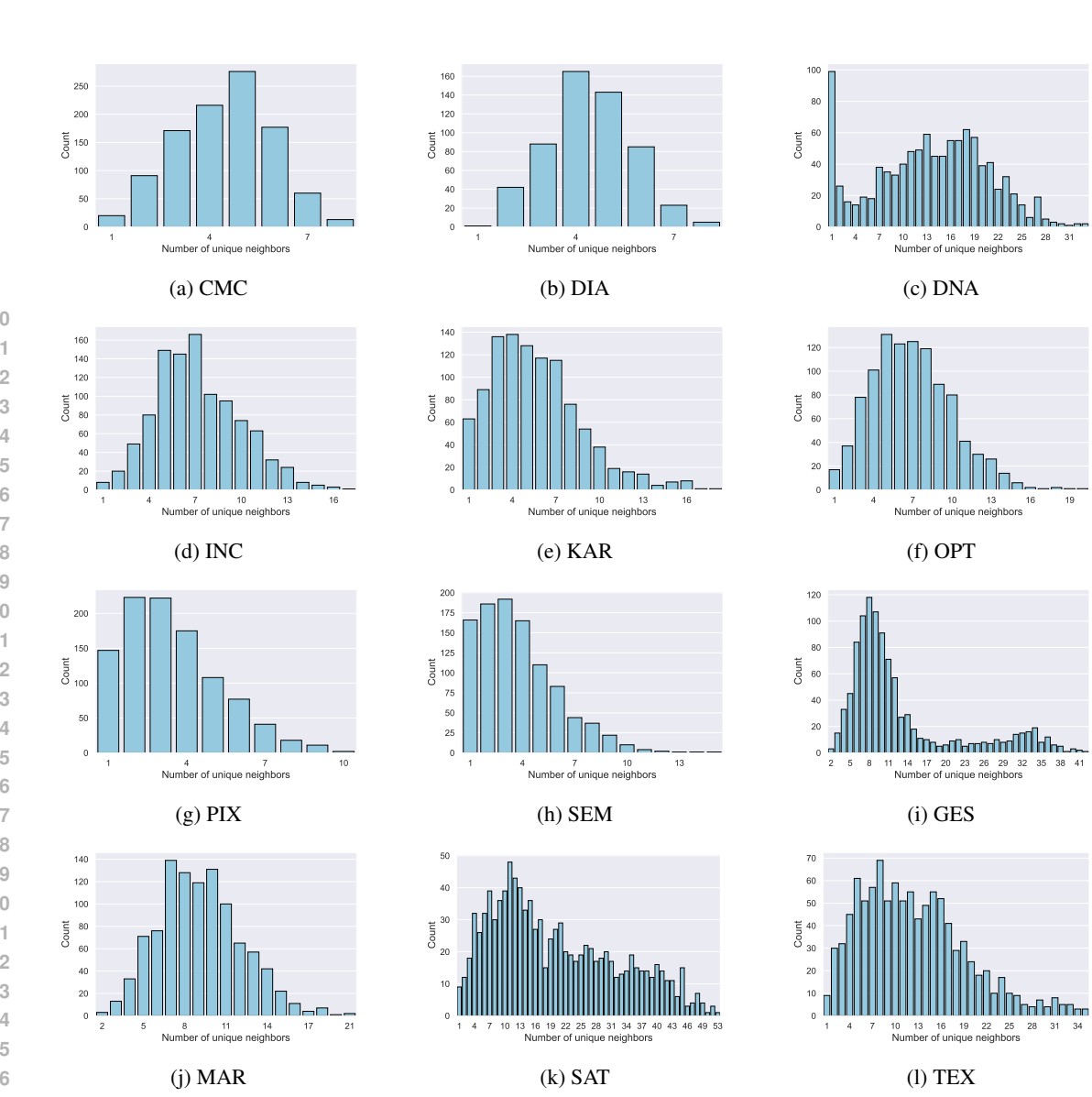

Figure 8: A distribution of the number of unique neighbors matched to the original instance. Low values indicate high stability of SeBA.

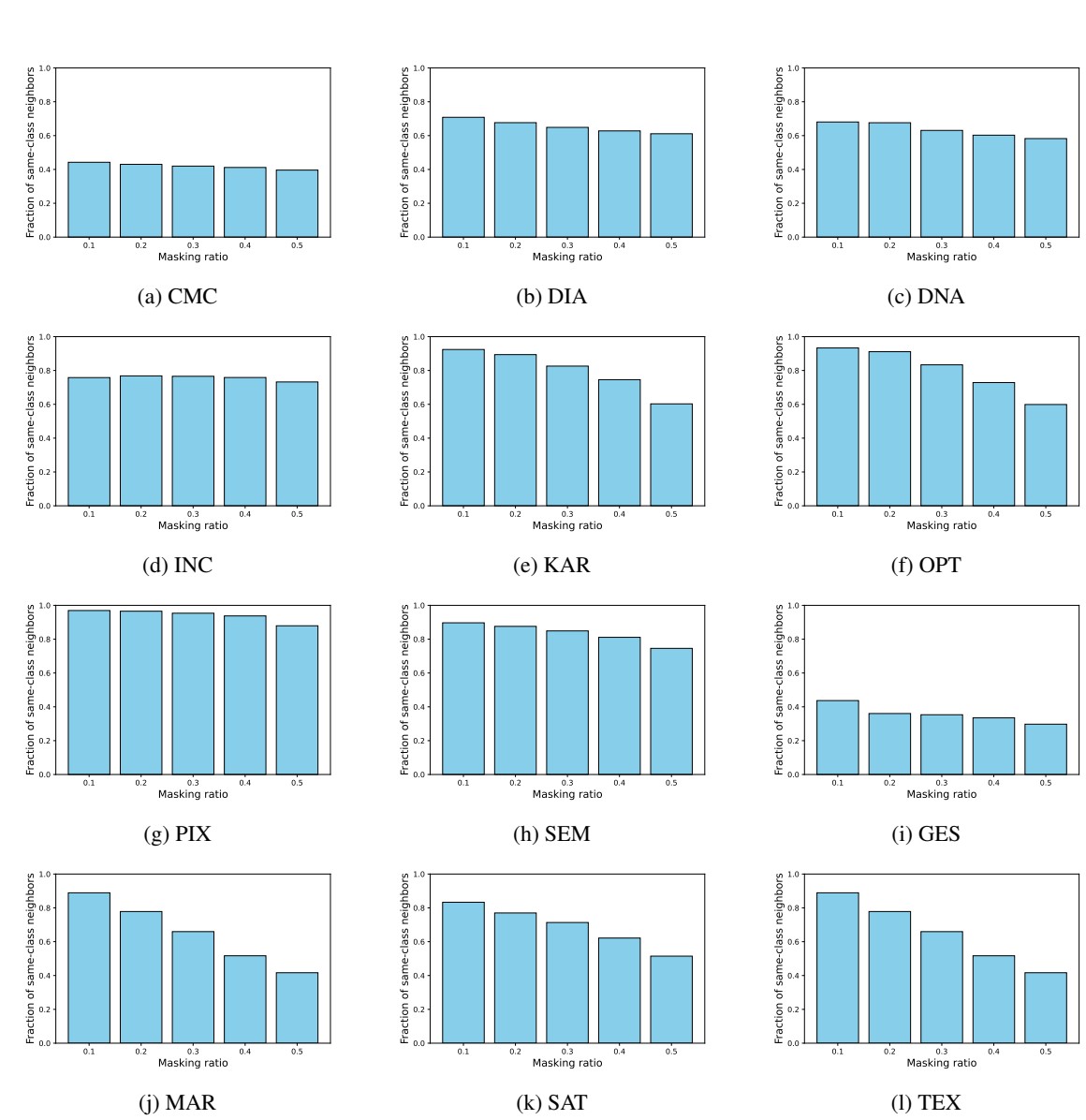

Figure 9: Relationship between masking ratio and the proportion of same-class nearest neighbors.

Table 10: Evaluation in terms of 1-, 5-, and 10-shot classification accuracy with standard deviations for 3 best-performing methods: STUNT (Nam et al., 2023) D2R2 (Liu et al., 2024), and SeBA. We put the best method in bold if it outperforms the second-best method by more than the standard deviation of its result. Out of 36 instances, there are 27 such cases and SeBA is the best-performing approach in 23 of them.

| Method | CMC | DIA | DNA | INC | KAR | OPT | PIX | SEM | GES | MAR | SAT | TEX |
|---|---|---|---|---|---|---|---|---|---|---|---|---|
| **#shot=1** | | | | | | | | | | | | |
| STUNT | 37.10 ±2.98 | 61.08 ±4.47 | 66.20 ±6.26 | 63.52 ±4.75 | 71.20 ±3.32 | 76.94 ±2.28 | 79.05 ±3.63 | 55.91 ±3.65 | 27.04 ±2.71 | 53.88 ±5.62 | 63.12 ±4.20 | 58.69 ±3.07 |
| D2R2-c | **40.81** ±0.41 | 60.10 ±2.11 | 61.29 ±2.16 | **72.85** ±2.03 | 61.45 ±1.78 | 77.41 ±1.56 | 61.45 ±1.15 | 34.26 ±1.63 | 26.58 ±0.44 | 51.70 ±3.51 | 60.92 ±1.25 | 61.78 ±1.44 |
| SeBA (our) | 36.76 ±0.72 | 61.14 ±1.15 | 66.79 ±2.62 | 62.89 ±1.87 | **76.40** ±1.01 | **78.94** ±0.88 | **83.06** ±1.20 | **61.11** ±1.07 | 27.07 ±0.69 | **58.43** ±3.77 | **65.70** ±0.88 | **70.94** ±0.78 |
| **#shot=5** | | | | | | | | | | | | |
| STUNT | 40.40 ±3.55 | 69.88 ±7.87 | 79.18 ±6.54 | 72.69 ±4.31 | 85.45 ±2.11 | 88.42 ±1.97 | 89.08 ±2.06 | 71.54 ±3.00 | 32.19 ±1.85 | 58.62 ±5.52 | 74.25 ±2.45 | 68.57 ±2.57 |
| D2R2-c | **43.39** ±0.37 | 68.69 ±1.63 | **81.39** ±1.38 | 73.34 ±1.91 | 79.49 ±0.95 | 87.12 ±0.91 | 82.22 ±1.93 | 60.16 ±1.42 | 30.26 ±0.25 | 56.24 ±1.75 | 70.66 ±0.63 | 71.82 ±0.60 |
| SeBA (our) | 42.85 ±0.62 | 69.54 ±0.71 | 79.86 ±2.45 | 71.28 ±0.65 | **87.59** ±0.47 | **90.11** ±0.45 | **91.88** ±0.88 | **79.41** ±0.77 | 32.07 ±0.50 | **65.22** ±2.51 | **78.66** ±0.38 | **87.51** ±0.47 |
| **#shot=10** | | | | | | | | | | | | |
| STUNT | 42.01 ±5.18 | 72.82 ±4.17 | 80.96 ±4.92 | 74.08 ±3.33 | 86.95 ±2.49 | 89.91 ±1.40 | 89.98 ±2.44 | 74.74 ±3.08 | 34.30 ±2.43 | 61.08 ±7.63 | 75.58 ±1.69 | 71.10 ±1.49 |
| D2R2-c | 37.86 ±0.58 | 72.02 ±1.61 | 81.72 ±1.89 | 75.34 ±1.97 | 84.81 ±0.75 | 89.27 ±0.74 | 73.70 ±1.17 | 36.30 ±1.28 | 31.22 ±0.19 | 59.80 ±1.38 | 71.96 ±0.56 | 73.43 ±0.51 |
| SeBA (our) | **46.30** ±0.62 | **73.61** ±0.43 | 83.59 ±2.21 | 72.68 ±0.52 | **90.88** ±0.38 | **92.62** ±0.42 | **93.88** ±0.43 | **84.11** ±0.74 | 34.60 ±0.37 | **69.96** ±1.87 | **81.17** ±0.23 | **90.18** ±0.36 |

