# OpenReview forum: "SeBA: Semi-supervised few-shot learning via Separated-at-Birth Alignment for tabular data"
_ICLR.cc/2026/Conference — Submitted to ICLR 2026_

### Official Review · Reviewer_LvJK · 2025-10-23

**Soundness:** 2
**Presentation:** 3
**Contribution:** 2
**Rating:** 4
**Confidence:** 5

**Summary:**

The paper presents SeBA, a novel augmentation-free SSL framework for tabular data. It splits each sample into two complementary views — feature and target — and learns to align the nearest neighbor relationships identified in the target view within the representation space of the feature view, under the assumption that tabular features are redundant and semantically correlated, allowing partial views to preserve the same underlying structure.

**Strengths:**

- The paper is easy to follow.
- The paper is well motivated, addressing the challenge of meaningful augmentation in tabular learning.
- The paper introduces a novel augmentation-free SSL framework, which based on JEAs, for tabular data.
- The framework is conceptually simple yet generalizable.

**Weaknesses:**

- The paper lacks evaluation fairness across datasets. While SeBA adds new datasets, it remains unclear under what criteria these datasets were added and how representative they are of broader tabular domains. Several entries in Tables 1–2 remain blank, leaving unclear whether SeBA’s advantage stems from better representation learning or from dataset selection bias.
- The paper lacks organization and comprehensive baseline coverage. The evaluation does not clearly separate supervised, semi-supervised, and self-supervised baselines, resulting in a fragmented comparison. Several widely used methods are omitted: classical semi-supervised algorithms such as Pseudo-Labeling [1], Mean Teacher [2], and ICT [3], as well as strong tabular SSL frameworks like SAINT [4]. Moreover, recent contrastive and range-limited augmentation approaches (e.g., FESTA [5]) directly address the same challenge of constructing semantically valid augmentations for tabular data. Including and organizing these baselines would provide a fairer and more interpretable evaluation of SeBA’s contribution within the broader SSL landscape.
- The paper shows an unreliable alignment assumption on certain datasets, as the core premise—that samples close in the target-view space are semantically similar—does not hold for datasets such as CMC and GES, where feature redundancy is low. This suggests that the alignment mechanism is not reliable and becomes unstable when the underlying feature correlation is weak or when the domain structure is heterogeneous.
- The paper shows limited performance without ratio ensemble, as training with a single separation ratio 𝑇 leads to highly variable results across datasets. This suggests that the learned representations are mask-dependent rather than semantically invariant. The ratio ensemble compensates for this instability but requires training multiple encoders (≈5× compute), reducing scalability and reproducibility. More efficient alternatives—such as Dynamic Ratio Sampling or a ratio-conditioned encoder—could replace the multi-encoder ensemble by learning to handle varying separation ratios within a single network.

*Reference*

[1] Dong-Hyun Lee, Pseudo-label: The simple and efficient semi-supervised learning method
for deep neural networks. (ICML 13)

[2] Antti Tarvainen and Harri Valpola, Mean teachers are better role models: Weight-averaged consistency targets improve semi-supervised deep learning results. (NeurIPS 17)

[3] Vikas Verma et al., Interpolation consistency training for semi-supervised learning. (Neural Networks 22)

[4] Gowthami Somepalli et al., Saint: Improved neural networks for tabular data via row attention and contrastive pre-training. (TRL workshop at NeurIPS 22)

[5] K Lee et al., Range-limited Augmentation for Few-shot Learning in Tabular Data with Comprehensive Benchmark. (KDD 25)

**Questions:**

[Q1] Dataset selection criteria

What were the criteria for adding the new datasets from OpenML-CC18 (e.g., domain diversity, feature-correlation profiles, class balance)?
Additionally, could the authors clarify why the results (“–” entries in Tables 1–2) for these datasets were left unreported, and whether they could be provided to ensure evaluation fairness and reproducibility?

[Q2] Baseline coverage

Would additional comparisons with well-known semi-supervised methods (Pseudo-Labeling [1], Mean Teacher [2], ICT [3]), strong tabular SSL models such as SAINT [4], and recent contrastive or range-limited augmentation approaches like FESTA [5] provide further insight into SeBA’s strengths? Such comparisons could better demonstrate how SeBA differs from and potentially improves upon existing paradigms that also address the challenge of defining semantically valid augmentations in tabular data.

[Q3] Alignment stability and assumption validity

In datasets like CMC and GES, where feature redundancy is low, the alignment assumption seems unreliable.
Have the authors analyzed whether the instability arises from the neighbor selection process or from dataset-specific feature sparsity?
Would a feature-group-aware masking strategy mitigate this issue?

[Q4] Dependence on ratio ensemble

Could the authors clarify whether the ratio ensemble is a core design choice that contributes to SeBA’s effectiveness, or merely a compensatory mechanism to mitigate instability in single-ratio training?
Have the authors explored a single-model variant, for example by sampling different separation ratios 𝑇  for each batch (Dynamic Ratio Sampling) or by explicitly conditioning the encoder on the current ratio (ratio-conditioned encoder), so that one model learns to handle multiple ratio distributions without ensembling?

---

> ### Author Response · Authors · 2025-11-22
> **Response to Reviewer LvJK**
>
> We thank Reviewer LvJK for the thorough feedback. We expanded the evaluation, added several new baselines, clarified dataset selection, and provided new analyses to fully address the concerns raised.
> All revisions are marked in blue in the updated manuscript. We address each weakness (W) and question (Q) below:
>
> ### W1 / Q1: Results on GES, MAR, SAT and TEX datasets
>
> We added GES, MAR, SAT, and TEX because they are standard components of the OpenML-CC18 benchmark, chosen for their diversity in domain, dimensionality, and class imbalance (e.g., MAR is strongly imbalanced). This yields a more representative evaluation of modern tabular tasks.
> In the revision, we report all remaining baselines on these datasets (Tables 1-2).  The UMTRA, SES and CACTUS approaches were neither designed nor adapted for tabular data in their publicly available implementations. We re-report these results from the STUNT paper,  whose authors performed such adaptation on their own. TabPFN cannot be evaluated on TEX because its public implementation supports ≤10 classes, while TEX has 11. SEBA remains the generally best-performing approach  across the expanded benchmark.
>
>
>
> ### W2 / W3 / Q2: Baseline coverage
>
> We expanded the baseline coverage in the revised submission by adding T-JEPA, Pseudo-Labeling, and Mean Teacher to the 1-shot and 5-shot comparisons (Tables 1-2).  These additions cover the main methodological families highlighted in the review and allow a representative evaluation of semi-supervised and augmentation-free approaches in the tabular SS-FSL setting. Across all newly added baselines, SEBA continues to match or outperform existing methods, reinforcing the differences between SeBA and prior work. As already noted in Appendix A.2, we group all methods into four clear categories (supervised, self-supervised, meta-learning, and semi-supervised), ensuring an organized and comprehensive comparison.
>
> ### W4: Reliability of the alignment mechanism
>
> Empirical evaluation, presented in Fig 3a, confirms that SeBA objective aligns with the downstream classification task -- in a great majority of cases, the nearest neighbors found in a target view share the same class as the original instance (in the next point, we answer the question regarding CMC and GES datasets).
> To analyze sensitivity, to k, we provide a variant of Analysis presented in Fig. 3a where the label agreement is not defined in terms of a single, but k nearest neighbors. The newly added Fig. 3b and Fig. 7 show that the immediate nearest neighbor provides the highest alignment with the target task among top k neighbors.
>
> ### W4/Q3: Low performance on CMC and GES datasets
>
> We verified that the accuracy of an MLP classifier trained on the fully-labeled CMC dataset equals 48%, which is only 2 percentage points higher than SeBA obtains in a 10-shot setting. CMC contains 9 features, and most of them are categorical. We verified that 573 samples (out of 1473 samples) have more than one nearest neighbor with an identical distance. This makes the classification task on this dataset so difficult.
> GES is composed of features extracted from videos with people gesticulating (features describe velocity and acceleration of hands and wrists). In consequence, this dataset has little connection with a typical tabular domain, which may suggest why our NN alignment procedure does not deliver meaningful positive pairs.
>
> ### W5 / Q4: Dependency on masking ratio
>
> The ratio ensemble is not a core requirement of SEBA, but it provides a simple and robust way to avoid dataset-specific tuning; an aspect highlighted as a strength by Reviewers iSr3 and nyXb. Practitioners may of course select a single ratio tailored to their dataset, but ensembling offers a practical default that performs well across diverse settings. We also explored dynamic ratio sampling within a single model; however, its performance was less stable than the multi-ratio ensemble (see Table 7), supporting our choice of ensemble as the most reliable strategy in practice.

---

> > ### Comment · Reviewer_LvJK · 2025-11-24
> >
> > Dear Authors,
> >
> > Thank you for detailed response to my questions. However, there are still a few concerns that remain unresolved, and I would like to follow up on them for further clarification.
> >
> > ---
> >
> > **[Q1] Results on GES, MAR, SAT and TEX datasets**
> >
> > I am still confused about the explanation regarding UMTRA, SES, and CACTUS. If these methods are “not designed or adapted for tabular data,” then it is unclear why they produce results on most of the tabular datasets in Tables 1–2, but not on GES, MAR, SAT, and TEX specifically. All of these datasets are tabular. Therefore, I do not understand why tabular incompatibility would affect only this subset of datasets.
> >
> > Moreover, since the STUNT paper already adapted these methods for tabular data and reported results using that adapted implementation, it seems that SeBA should be able to reproduce the same procedure to obtain results on all tabular datasets, including the new ones. If STUNT’s adaptation was used for the other datasets, why was the same approach not applied to GES, MAR, SAT, and TEX as well? If there is a valid reason for this discrepancy, it should be clearly stated in the main paper.
> >
> > And based on your response to Q3, given that GES is described as “not a typical tabular dataset” and is inherently unsuitable for SeBA’s alignment mechanism, could the authors clarify why it was added to the benchmark at all?
> > And since FESTA evaluates 42 OpenML datasets, why did SeBA include only these four?
> >
> > ---
> >
> > **[Q2] Baseline coverage**
> >
> > I appreciate the additional baselines that were included.
> > However, several key methods I previously mentioned—such as SAINT, ICT, and FESTA—are still missing from the comparison, and the reason for their exclusion is not fully explained in the response.
> >
> > These methods are not minor variants; they represent important families of approaches directly relevant to the SSL/SS-FSL setting or the challenge of defining semantically meaningful augmentations for tabular data. Without them, it remains unclear how SeBA compares against the broader landscape of existing methods, especially those that address similar challenges.
> >
> > Could the authors clarify why these baselines were not included or provide justification for their exclusion?
> > If possible, adding at least some of these methods would substantially strengthen the empirical evaluation.
> >
> > ---
> >
> > **[Q4] Dependence on ratio ensemble**
> >
> > Given that the ratio ensemble is stated to be “not a core requirement,” the single-ratio results become essential for evaluating the mechanism itself. Figure 4b shows that single-ratio SeBA models have substantially lower and unstable accuracy across datasets. Because tuning the ratio is part of standard model selection, I would like to request that the authors.
> >
> > (1) A validation-based tuning of T,
> >
> > (2) Followed by reporting the absolute test performance of the tuned single-T model on the full benchmark,
> >
> > (3) Compared directly to the baselines in Tables 1–3.
> >
> > Without these results, it remains unclear whether SeBA’s improvements stem from the proposed "Separated-at-Birth Alignment" itself or from the multi-ratio ensemble. Since the ensemble is claimed to be non-core, the single-model results are necessary to evaluate the method’s primary contribution.
> >
> > Furthermore, if my understanding is correct, the current formulation of SeBA requires training k independent encoders—one for each masking ratio 𝑇∈{𝑇_1,…,𝑇_𝑘}—and aggregating their predictions at inference time. This effectively multiplies both training and inference cost by a factor of k, which substantially complicates the practical use of SeBA and further motivates the need to verify whether a single, properly tuned model can achieve competitive performance without relying on the ensemble.

---

> > > ### Author Response · Authors · 2025-11-25
> > > **Response to Reviewer LvJK**
> > >
> > > We thank the reviewer for their follow-up and detailed comments.
> > >
> > > ### [Q1] Results on GES, MAR, SAT and TEX datasets for UMTRA, SES and CACTUs.
> > >
> > > We would like to clarify that the official repositories of UMTRA, SES, and CACTUs implement these methods exclusively for vision-based meta-learning (few-shot image classification), without tabular adaptations. Reproducing these approaches in the tabular setting therefore requires unspecified procedural steps that are not described in the original works. The STUNT paper reports internal tabular variants, but these adaptations are not released, and Appendix I of STUNT does not  specify the procedural steps required to reproduce these variants in a controlled manner. Any reimplementation would therefore introduce author-dependent choices, leading to non-comparable outcomes. For this reason, we focus on methods with established tabular pipelines (SubTab, VIME, SCARF, T-JEPA) and widely-used semi-supervised methods (Pseudo-Labeling, Mean Teacher), which allow a principled and reproducible SS-FSL comparison.
> > > As for the GES dataset, while it is not a typical tabular dataset, it has been prominent in multiple tabular benchmarks. Given its particular challenging nature, it is important to monitor and report the performance of emerging SS-FSL approaches on this dataset.
> > >
> > > ### [Q2] Baseline coverage
> > >
> > > Thank you for the suggestion. We expanded the baseline coverage in the revised submission by adding T-JEPA, Pseudo-Labeling, and Mean Teacher (Tables 1-2), which represent the main methodological families relevant to SS-FSL and tabular pretraining. Across these additions, SeBA continues to match or outperform existing methods.
> > >
> > > To our knowledge, FESTA does not provide a publicly available official implementation, and ICT provides an official implementation suitable only for image-based CIFAR and SVHN datasets. Reproducing or adapting them for tabular data would require making multiple non-trivial unspecified implementation decisions, resulting in non-unique reproductions and undermining a fair and controlled comparison. We therefore restrict our evaluation to methods with established and reproducible tabular pipelines.
> > >
> > > SAINT is a relevant representative of deep tabular models. We have begun running SAINT under the same SS-FSL protocol as the other baselines  to ensure parity of evaluation. Given the computational cost of its training, this evaluation is ongoing; we will share the results during the discussion.
> > >
> > > ### [Q4] Dependence on ratio ensemble
> > >
> > > In realistic SS-FSL settings, supervised hyperparameter selection on a validation set is infeasible; the support set typically contains only 5-10 labeled examples, and partitioning them into train/validation folds reduces the already limited supervisory signal and weakens the final model. Accordingly, SeBA emphasizes robustness at pretraining time rather than label-driven hyperparameter tuning.
> > >
> > > Regarding computational cost, SeBA ensembles consist of five lightweight 2-layer MLP encoders (~0.59M FLOPs each), totaling ~2.9M FLOPs per forward pass. By comparison, D2R2, a strong baseline in the same SS-FSL setting, exceeds 5M FLOPs per iteration due to dual encoders, diffusion noise prediction, and distance-matching, even before backpropagation. Thus, SeBA achieves comparable performance without the computational overhead of such approaches.
> > > The masking ratio is the only SeBA hyperparameter, which substantially narrows the configuration space. This stands in contrast to methods such as D2R2, which require multi-dimensional hyperparameter grid-searches across architecture, diffusion schedule, and augmentation settings.
> > >
> > > Finally, to address the reviewer’s remark, we  note that in practice, a single-encoder variant of SEBA with a fixed separation ratio of 0.3 also provides competitive performance across datasets without any dataset-specific tuning.  The ensemble serves primarily to reduce variance in extreme few-shot settings.

---

> > > > ### Author Response · Authors · 2025-11-27
> > > > **Response to Reviewer LvJK - updated SAINT results**
> > > >
> > > > We thank the reviewer for their comments.
> > > >
> > > > As indicated previously, we have updated the revised manuscript with SAINT results under the same SS-FSL protocol used for all baselines. SAINT performs strongly on GES, while across the remaining datasets, SeBA achieves stronger performance while using a substantially lighter architecture and no dataset-specific tuning. This is consistent with our stated goal of stable SS-FSL performance under minimal assumptions and resource constraints.

---

### Official Review · Reviewer_Jx13 · 2025-10-26

**Soundness:** 3
**Presentation:** 3
**Contribution:** 2
**Rating:** 4
**Confidence:** 4

**Summary:**

This paper proposes Separated-at-Birth Alignment (SeBA), a semi-supervised few-shot learning (SS-FSL) framework specifically designed for tabular data. The SeBA addresses the challenge of learning representations from scarce labeled samples with access to many unlabeled ones.

The key innovation is to remove the augmentation dependency that is typical in self-supervised learning (SSL). Instead of generating positive pairs through data augmentations, SeBA divides the input features into two complementary “views” (a feature view and a target view) and aligns the learned representations of the feature view with the nearest-neighbor graph derived from the target view.
The method introduces a type-aware separation scheme to handle mixed categorical and numerical attributes, ensuring semantic consistency when creating the two views.
SeBA uses a lightweight multi-layer perceptron (MLP) encoder and a conditioned projector, combined with an ensemble strategy across multiple separation ratios to improve generalization and avoid overfitting on small datasets.

**Strengths:**

1. The paper addresses an important gap in the literature. Few-shot and semi-supervised learning have been well explored in vision and NLP, but the tabular domain remains underdeveloped. SeBA offers a principled approach to this problem.

2. The method eliminates the need for artificial augmentations. This is a substantial conceptual improvement, as data augmentations are ill-defined or even harmful in tabular data, and SeBA provides a clear alternative.

3. The design is elegant and simple. By constructing “separated-at-birth” views and aligning representations via nearest neighbors, the framework remains lightweight yet effective.

**Weaknesses:**

1. The paper’s scope is somewhat narrow. Although the title and framing emphasize “semi-supervised few-shot learning,” all experiments are limited to tabular data. The contribution might not generalize to other modalities such as vision or multimodal data.

2. The novelty is incremental within the self-supervised learning paradigm. The core idea is constructing paired views without augmentation. This idea is conceptually related to existing joint-embedding methods (e.g., BYOL, SimCLR) with different pairing mechanisms.

3. The method relies on the assumption of meaningful nearest neighbors. The nearest-neighbor graph in the target view may not always reflect true semantic relationships, especially in high-noise or high-dimensional data.

4. The evaluation lacks direct comparison to transformer-based tabular models. Models like TabPFN and UniTabE are mentioned but not deeply analyzed in the few-shot regime, which could provide a more rigorous benchmark.

5. In recent years, Multi-modal Large Language Models (MLLMs) have demonstrated strong capabilities, featuring large parameter sizes and excellent generalization. In contrast, the model proposed in this paper appears to have a relatively small number of parameters. I believe that the proposed method could potentially be integrated into MLLM frameworks, but the authors have not explored this direction.

**Questions:**

Please address the concerns I raised in the Weaknesses section.

In addition, could the authors include qualitative examples of the datasets and model outputs in the main text (or in the supplementary material)?

**Details Of Ethics Concerns:**

None.

---

> ### Author Response · Authors · 2025-11-22
> **Response to Reviewer Jx13**
>
> We thank Reviewer Jx13 for the insightful comments. We have addressed all concerns through additional explanations, theoretical and empirical support, and expanded comparisons in the revised version.
> All revisions are marked in blue in the updated manuscript. We address each weakness (W) below:
>
> ### W1: Scope of the paper
>
> We agree that our focus is on tabular data, but we do not view this as a narrow scope. Tabular data is one of the most prevalent modalities in real-world ML (e.g., healthcare, finance, scientific datasets) and presents unique challenges, like the lack of natural augmentation, that do not arise in text or vision. Our contribution is therefore intentionally domain-specific, addressing a problem space that is both broad in practice and insufficiently explored in the literature.
>
> ### W2: Incremental novelty within SSL paradigm
>
> We agree that our work fits within the joint-embedding (J-E) representation-learning paradigm, but the key challenge we address is that standard J-E methods rely on meaningful augmentations - an assumption that holds in vision but fundamentally breaks down for tabular data. This is precisely why prior tabular J-E approaches such as SCARF and SubTab produce significantly weaker representations than current SOTA methods like D2R2 and STUNT. Our contribution is to show that separation-at-birth provides a principled way to construct positive pairs without augmentations, enabling J-E architectures to reach substantially higher representation quality on tabular data. This makes J-E methods practical and competitive in the challenging tabular domain where they previously underperformed.
>
> ### W3: Theoretical justification of the assumption of meaningful nearest neighbors
>
> To support empirical analysis of nearest neighbor alignment with a target task (Fig 3a & 6), we conducted a theoretical analysis taking into account classes following Gaussian distributions, see the newly added Appendix C for details. We derive that the expected value of the probability of the mismatch event (a sample and its nearest neighbor calculated in a target view come from different classes) decays exponentially as the distance between Gaussian classes increases in the original space.  Moreover, the probability of this event gets lower as the target views contain more of the original features. That is why we use masking ratios in the range [0.1, 0.5], which means that we discard at most half of the original features to compute nearest neighbors. Concluding, if the classes are separated well enough, then pretext tasks generated by SeBA are informative for a target task.
>
> ### W3: Empirical justification of the assumption of meaningful nearest neighbors
>
> Empirical evaluation, presented in Fig 3a, confirms that SeBA objective aligns with the downstream classification task -- in a great majority of cases, the nearest neighbors found in a target view share the same class as the original instance.
> To analyze sensitivity, to k, we provide a variant of Analysis presented in Fig. 3a where the label agreement is not defined in terms of a single, but k nearest neighbors. The newly added Fig. 3b and Fig. 7 show that the immediate nearest neighbor provides the highest alignment with the target task among top k neighbors.
>
>
> ### W4: Comparison to Transformer architectures
>
> We compare SEBA to TabPFN and T-JEPA, recent prominent transformer-based tabular foundation models, and find that SEBA consistently achieves superior performance. We have also added the performance results of these models on all datasets, which further confirm this trend (see Tables 1-2). To our knowledge, UniTabE does not provide a publicly available codebase, preventing a fair and reproducible comparison. To isolate the effect of encoder architecture, we additionally compare SEBA using an MLP encoder versus an FT-Transformer encoder (see Table 9). The MLP achieves higher accuracy in the vast majority of cases while being substantially more computationally efficient, supporting its suitability for tabular few-shot learning.
>
>
> ### W5: Integration into MLLM frameworks
>
> We thank the reviewer for noting that SEBA could be integrated into larger multimodal models, which we view as a positive indication that the representations learned by our approach are broadly useful. While exploring such integration is beyond the scope of this work, which focuses on an efficient and principled solution for tabular SS-FSL, we agree that SEBA’s simplicity and strong representations make it a promising candidate for future incorporation into MLLM frameworks.

---

### Official Review · Reviewer_nyXb · 2025-10-30

**Soundness:** 3
**Presentation:** 3
**Contribution:** 2
**Rating:** 6
**Confidence:** 3

**Summary:**

The authors address the challenge of semi-supervised few-shot learning (SS-FSL) for tabular data, where models must classify data with very limited labeled examples while leveraging a large pool of unlabeled data. This problem is particularly relevant in domains like medical diagnosis, credit risk prediction, and cognitive sciences where obtaining labeled data is expensive but unlabeled data is readily available. The research aims to develop an effective pretraining method specifically tailored for tabular data that can learn meaningful representations from unlabeled data and then be fine-tuned with just a few labeled examples.

The paper identifies critical limitations of existing self-supervised learning (SSL) methods when applied to tabular data. Traditional SSL approaches rely heavily on data augmentations to create semantically similar positive pairs for contrastive learning. While augmentations like cropping, rotation, or color jittering work naturally for images, defining meaningful augmentations for tabular data is problematic. Poorly chosen transformations such as zero masking, Gaussian noise, or sampling from marginal distributions can distort semantic meaning, generate out-of-distribution samples, or create invalid data points. For instance, decreasing a car's age while increasing its mileage would be semantically inconsistent, or assigning non-integer values to discrete features like number of car seats would be invalid. This fundamental challenge has led recent work to largely abandon SSL for tabular data in favor of alternative approaches like cluster detection or diffusion-based methods.

SeBA introduces a novel approach that eliminates the need for augmentations entirely. The core idea involves separating tabular data "at birth" into two complementary and independent views: feature views and target views. For each minibatch, a random binary mask determines which columns belong to each view. The method then identifies nearest-neighbor relationships in the target view space and trains an encoder to align the representations of feature views according to these nearest-neighbor correspondences.
The authors argue this works well for several reasons. First, it avoids the problematic task of designing augmentations for tabular data. Second, the nearest-neighbor relationships provide semantically meaningful positive pairs based on actual data similarity rather than artificial transformations. Third, the method employs a conditioned projector that takes both the encoder representation and the separation mask as inputs, allowing the model to adapt to different separation schemes. Fourth, type-aware separation ensures categorical variables are handled properly without splitting their one-hot encodings. Finally, an ensemble strategy using multiple separation ratios eliminates the need for careful hyperparameter tuning.

The authors conduct extensive experiments across twelve tabular datasets in 1-shot, 5-shot, and 10-shot classification settings. They compare SeBA against multiple baseline categories including supervised methods like CatBoost and k-NN, self-supervised methods like VIME and SCARF, meta-learning approaches, and state-of-the-art SS-FSL methods STUNT and D2R2. Performance is measured through classification accuracy with multiple random seeds and support/query set selections to ensure statistical reliability.
Additionally, the authors provide detailed ablation studies examining the impact of data normalization, missing data imputation strategies, separation ratios, and classifier choices. They also analyze the alignment between the pretraining objective and downstream tasks by measuring the proportion of nearest neighbors that share the same class label and examining the stability of neighbor assignments across different random separations.

The authors conclude that SeBA successfully demonstrates that self-supervised learning can be effective for tabular data when properly designed. The method achieves state-of-the-art performance on tabular few-shot learning benchmarks, obtaining the best accuracy in 29 out of 36 experimental instances. The main contributions include introducing the Separated-at-Birth Alignment framework that eliminates augmentation requirements, instantiating it as a lightweight model with ensemble strategies to prevent overfitting, providing thorough empirical validation, and demonstrating consistent generalization across diverse tabular datasets. The work opens new avenues for SSL paradigms in tabular data and provides a practical foundation for data-constrained applications.

**Strengths:**

SeBA addresses the fundamental incompatibility between traditional SSL and tabular data by completely reimagining how positive pairs are constructed. Rather than forcing unnatural augmentations onto tabular data, it leverages the inherent structure of the data itself through nearest-neighbor relationships. This approach is particularly well-suited because tabular data naturally contains meaningful similarity structures that can be discovered through partial feature comparisons.

The key contribution lies in demonstrating that SSL principles can be successfully adapted to tabular data without relying on augmentations. By introducing the separated-at-birth concept, the authors provide a principled alternative to augmentation that maintains the benefits of contrastive learning while respecting the unique characteristics of tabular data. The type-aware separation scheme for handling mixed categorical and numerical features represents an important technical innovation that ensures semantic validity.

The method builds on solid theoretical foundations from contrastive learning while addressing specific tabular data challenges systematically. The use of InfoNCE loss provides a well-understood optimization objective, while the conditioned projector allows the model to handle varying separation schemes coherently. The ensemble approach addresses the practical challenge of hyperparameter selection in few-shot scenarios where validation data is scarce. Each design choice, from zero imputation to type-aware separation, is motivated by specific tabular data characteristics and supported by ablation studies.

**Weaknesses:**

The paper lacks theoretical justification for why nearest-neighbor relationships in partial feature spaces should consistently produce semantically meaningful positive pairs. While empirical results show high same-class neighbor rates, the conditions under which this assumption holds or might fail are not thoroughly analyzed. The relationship between separation ratio and dataset characteristics remains underexplored, leaving practitioners without clear guidance on when certain ratios might be preferred.

While the experiments cover multiple datasets and shot settings, certain aspects lack depth. The comparison with D2R2 uses only the inductive variant rather than the full transductive version that achieves better performance. The paper does not explore performance on datasets with very high dimensionality or extreme class imbalance, both common in real-world tabular applications. Additionally, computational efficiency comparisons are absent, which is important given the ensemble strategy requires training multiple models.

The ensemble approach, while eliminating hyperparameter tuning, significantly increases computational cost during both training and inference. The method's reliance on nearest-neighbor relationships may be problematic for datasets where local similarity doesn't align well with class structure, such as data with multimodal class distributions. The fixed separation ratios used in the ensemble might not be optimal for all datasets, particularly those with very few or very many features. Finally, the approach assumes that partial views contain sufficient information for meaningful nearest-neighbor matching, which might not hold for datasets with complex feature dependencies.

**Questions:**

Regarding the architectural choice, what is the theoretical advantage of using a lightweight MLP encoder over a transformer-based model for representation learning in the tabular SS-FSL setting, aside from the general benefit of avoiding overfitting on small datasets?

Can the authors clarify the underlying logical reason why zero imputation for masked features performs best in the SeBA framework compared to previous methods like sampling from empirical distribution, especially considering how this choice impacts the subsequent nearest-neighbor calculation in the target view?

Given the acknowledged weakness of SeBA's performance on the CMC and GES datasets, what specific characteristics of the data in these two datasets, such as dimensionality, feature distribution, or type complexity, might be hypothesized as the cause of the reduced efficacy of the Separated-at-Birth Alignment mechanism?

---

> ### Author Response · Authors · 2025-11-22
> **Response to Reviewer nyXb [1/2]**
>
> We thank Reviewer nyXb for the positive assessment and thoughtful suggestions. We have added further comparisons, analyses, and clarifications to strengthen the paper along the directions highlighted.
> All revisions are marked in blue in the updated manuscript. We address each weakness (W) and question (Q) below:
>
> ### Q1: Advantages of using an MLP
>
> While transformer models can work well on tabular data, their main strength lies in modeling rich feature interactions, which typically requires abundant labels or large-scale pretraining. In tabular SS-FSL, where labeled data are scarce, a simpler MLP can provide more stable and data-efficient representations without relying on complex attention patterns that are difficult to learn from few labels. For this reason, prior SOTA approaches such as STUNT and D2R2 also adopt an MLP encoder.
> Empirically, we directly compare SEBA with an MLP encoder and an FT-Transformer encoder, and find that the MLP achieves superior SS-FSL performance (see the newly added Tab. 9).
>
> ### Q2: Zero imputation
>
> We clarify that the imputation strategy does not affect the NN calculation in the target view, because nearest neighbors are computed directly on the unmasked target-view features. The choice of imputation only concerns the feature view. Zero imputation introduces no artificial signal about the masked columns, ensuring that the encoder cannot exploit spurious correlations. In contrast, sampling from the empirical distribution inserts synthetic values that may resemble real features, thereby injecting misleading information into the feature view and harming alignment.
>
> ### Q3: NN-alignment in CMC and GES datasets
>
> We verified that the accuracy of an MLP classifier trained on the fully-labeled CMC dataset equals 48%, which is only 2 percentage points higher than SeBA obtains in a 10-shot setting. CMC contains 9 features, and most of them are categorical. We verified that 573 samples (out of 1473 samples) have more than one nearest neighbor with an identical distance. This makes the classification task on this dataset so difficult.
> GES is composed of features extracted from videos with people gesticulating (features describe velocity and acceleration of hands and wrists). In consequence, this dataset has little connection with a typical tabular domain, which may suggest why our NN alignment procedure does not deliver meaningful positive pairs.

---

> > ### Comment · Reviewer_nyXb · 2025-11-25
> >
> > Thank you for your detailed answers to my questions.
> > I was able to understand most of the questions I had.

---

> ### Author Response · Authors · 2025-11-22
> **Response to Reviewer nyXb [2/2]**
>
> ### W1: The relationship between separation ratio and dataset characteristics
>
> The separation ratio is an important parameter for generating positive pairs: the larger the dimension of the target view, the greater the chance of producing a semantically meaningful positive pair, see Fig. 3d & Fig. 9. However, if we have too high dimensionality of the target view, the model has to predict nearest neighbor relationships based on low dimensionality input (feature view), which can be too complex to train. Since an optimal proportion may depend on a given dataset, we proposed an ensemble strategy, in which we aggregate multiple models with various separation ratios. The ensemble strategy eliminates the need for selection of this hyperparameter, gives the best overall result, as well as it is computationally efficient because SeBA is a very lightweight model.
>
> ### W2: Lack of comparison with an inductive variant of D2R2
>
> Comparing the transductive variant of D2R2 to SEBA and the other baselines would be an apples-to-oranges comparison. The transductive variant refines the classifier using the test examples themselves, which is not allowed in a standard SS-FSL evaluation and cannot be assumed in practical deployment. For this reason, we follow prior work and restrict all methods to the more practically relevant inductive setting.
>
> ### W3: Performance on datasets with very high dimensionality or extreme class imbalance
>
> Our evaluation already includes highly imbalanced and high-dimensional datasets. For example, in the Marketing dataset, the most frequent class appears over 7x more often than the least frequent one. The Semeion dataset contains 256 features, which is comparable to some of the highest-dimensional datasets in the recently proposed TabArena benchmark (NeurIPS 2025). SeBA maintains strong performance on both of these datasets.
>
> ### W4: Computational efficiency
>
> While ensembles can be costly in some settings, ours consists of five lightweight 2-layer MLPs. For an input size of 32, a single model requires only \~0.59M FLOPs, so the entire 5-model ensemble amounts to 2.9M FLOPs. In contrast, one of our strongest baselines (D2R2) uses a 3-layer 512-wide MLP encoder (\~1.16M FLOPs) plus a separate 3-layer 512-wide diffusion-noise MLP (\~1.16M FLOPs), performs two encoder passes per iteration (\~2.32M FLOPs), and incorporates pairwise random-projection distance matching, which adds ~0.5-1.0M FLOPs per batch. As a result, a single D2R2 forward pass during training routinely exceeds \~5M FLOPs, even before backpropagation or diffusion-loss overhead. In comparison, the full SeBA ensemble (\~2.9M FLOPs) is substantially cheaper both in absolute terms and relative to D2R2’s per-iteration cost, and remains negligible on any modern CPU or GPU.
>
> ### W5: Complexity of the training ensembling approach
>
> Since our only hyperparameter is the masking ratio in range (0, 1), the possible hyperparameter space of SEBA is limited. This makes it viable to either search for an optimal setting for a given dataset, or ensemble several models trained with different ratios.
> On the other hand, methods such as D2R2 perform a hyperparameter grid-search over more complex combinations of several parameters, which is more computationally expensive.

---

> > ### Comment · Reviewer_nyXb · 2025-11-25
> >
> > Thank you for your detailed feedback on the areas I felt were lacking.
> > Anyway, it didn't seem to have had a significant impact on my score.

---

### Official Review · Reviewer_iSr3 · 2025-11-01

**Soundness:** 2
**Presentation:** 2
**Contribution:** 2
**Rating:** 4
**Confidence:** 4

**Summary:**

In the Tabular Semi-supervised Few-shot Learning (SS-FSL) setting, to address the fundamental difficulty that conventional Self-Supervised Learning (SSL) methods face in defining data augmentations, the authors propose a new joint-embedding architecture called Separated-at-Birth Alignment (SeBA). The method splits the data into a complementary Feature View and Target View, and achieves augmentation-free construction of positive pairs by aligning the representation of the Feature View to the k-nearest-neighbor relationships defined in the Target View. Combining a type-aware separation scheme with an ensemble strategy over various separation ratios ($T$), SeBA is reported to outperform existing state-of-the-art methods across diverse benchmark datasets.

**Strengths:**

**Augmentation-Free SSL Paradigm**: By sidestepping the long-standing challenge of designing semantically meaningful augmentations for tabular data, the authors introduce a new SSL pretext task that combines feature separation with nearest-neighbor matching. This has clear potential to steer research directions in the area.

**Robustness-Oriented Design**: Instead of manually tuning the optimal feature separation ratio ($T$), the approach ensembles encoders trained with multiple $T$ values to secure generalization and reliability. This is a practical strategy under few-shot constraints.

**Effective Ablations**: The analyses show that, for missing-data imputation, zero imputation outperforms marginal sampling by about 3–5 percentage points. They also confirm that linear probing is the most effective few-shot classifier, providing concrete justification for key design choices.

**Weaknesses:**

(Novelty)


The proposed idea is theoretically very similar to existing augmentation-free, feature-separation-based–based SSL methods (e.g., T-JEPA, Thimonier et al., 2025). In particular, generating subsets via random masking of tabular data and learning structure based on another subset overlaps with the core concept of T-JEPA.


The theoretical and empirical distinctions from the most closely related prior work, T-JEPA, are not clearly articulated .


(Technical Quality)


Lack of reproducibility and stability verification: Standard deviations are missing for all methods in the key results (Tables 1 and 2). Without them, one cannot assess statistical significance—crucial for evaluating reproducibility and performance stability in few-shot settings—representing a serious deficiency in technical rigor.


Incomplete and potentially unfair baseline comparisons: Results on datasets where important baselines were added (MAR, SAT, TEX, etc.) omit strong methods such as TabPFN, SCARF, and UMTRA, making it difficult to judge whether the proposed method generalizes broadly against up-to-date baselines (Tables 1 and 2). No clear technical rationale is provided for these omissions.

Questionable justification for random masking / NN alignment: In tabular data, random masking can discard information from critical features, and defining positive pairs via nearest neighbors assumes that local manifold similarity in high-dimensional/sparse spaces reflects global semantic similarity. This strong assumption (Section 3.1) lacks rigorous theoretical or empirical support.


Insufficient justification for the Conditional Projector: The design of the conditional projector π(h,m)\pi(h, m)π(h,m), which re-conditions the Feature-View encoding on the mask (Equation 5), appears logically unnecessary, and the paper provides limited analysis of its added value.


(Significance)


While the work aims to offer methodological insights toward addressing long-standing challenges in tabular SSL, the absence of statistical stability (missing STDs) and incomplete baseline coverage prevent an objective assessment of whether the paper meaningfully advances the field. (Insufficient evidence; reason: no statistical significance testing possible.)


(Writing & Presentation)


The overall structure and methodological exposition are clear. However, the omission of standard deviation information for the key experimental results (Tables 1 and 2) substantially limits readers’ ability to judge the reliability of the findings. Details necessary for reproducibility—especially those concerning statistical stability—are insufficient.

**Questions:**

Request for sensitivity analysis of NN alignment: To validate the methodology of using Nearest Neighbors (NN) as surrogates for positive pairs, please present an analysis of the final performance and the structural changes in the embedding space as the value of
𝑘 varies. In particular, could you quantitatively analyze how frequently semantic mismatch occurs when the definition of NN does not align semantically?

Request for explanation of missing baseline experiments: Please explain the specific reasons why the results for certain datasets (e.g., MAR, SAT, TEX) are missing for major baselines such as TabPFN, SCARF, and UMTRA in Tables 1 and 2. If possible, please provide additional experimental results on the missing datasets using the codebases of those baselines to ensure fairness in comparison.

Strengthening the distinction from T-JEPA: Please provide a detailed analysis of the theoretical and experimental differences between T-JEPA (Thimonier et al., 2025) and SeBA’s nearest-neighbor alignment–based approach. Clearly explain whether the two approaches pursue fundamentally different learning objectives rather than being simple variations of each other.

---

> ### Author Response · Authors · 2025-11-22
> **Response to Reviewer iSr3 [1/2]**
>
> We thank Reviewer iSr3 for the detailed and constructive feedback. We carefully addressed each point through additional analyses, expanded baselines, and clarifications in the revised manuscript.
> All revisions are marked in blue in the updated manuscript. We address each weakness (W) and question (Q) below:
>
> ### W1/Q3: Comparison to T-JEPA
>
> While both models involve separating the data into two masked views, there are several substantial differences between SEBA and T-JEPA.
>
> 1. The methods optimize different objectives:
> * T-JEPA: predicts the latent representation of missing features of the same sample - learns intra-sample feature dependencies.
> * SEBA: aligns inter-sample nearest-neighbor relationships - learns instance-to-instance similarity structure needed for few-shot classification.
>
> 2. The approaches make predictions in different spaces:
> * T-JEPA - latent space of the target model
> * SEBA: the NN relationships are computed in the target view space, i.e. a subspace of the data space, and not in any latent space.
>
> 3. The approaches avoid representation collapse differently:
> * T-JEPA: no negatives; relies on an empirically introduced “regularizer token” to avoid collapse.
> * SEBA: uses contrastive InfoNCE with negative pairs, preventing collapse by design.
>
> 4. There are different encoder architectures:
> * T-JEPA: uses transformers as encoder, decoder, and predictor modules, making both pretraining and inference more computationally complex.
> * SEBA: uses a lightweight 2-layer MLP as an encoder and an additional 2-layer MLP as a projector.
>
> Empirically, we compared the representations trained with T-JEPA on 1-shot and 5-shot classification (see Tables 1-2) and found T-JEPA to yield performance inferior to that of SEBA, further underlining the differences between the two methods.
>
>
> ### W2: Stability verification
>
> We would like to clarify that our experiments were run on 100 random seeds to ensure statistical robustness. In the revised paper, We added the standard deviations for the three strongest and most relevant methods in this setting (i.e. STUNT, D2R2, and SEBA) in Table 10. For the remaining baselines, results are taken from publicly available implementations or prior benchmark papers, which do not provide variance estimates or for which reproducible tabular SS-FSL code is not consistently available. Across the reproducible baselines, SEBA exhibits lower or comparable variance: in 27/36 cases, the best-performing method exceeds the second-best by more than one standard deviation, and SEBA is the best-performing method in 23 of these cases. This indicates that our conclusions are robust to statistical fluctuations.
>
> ### Q2/W3: Results on GES, MAR, SAT and TEX datasets
>
> We added GES, MAR, SAT, and TEX because they are standard components of the OpenML-CC18 benchmark, chosen for their diversity in domain, dimensionality, and class imbalance (e.g., MAR is strongly imbalanced). This yields a more representative evaluation of modern tabular tasks.
> In the revision, we report all remaining baselines on these datasets (Tables 1-2).  The UMTRA, SES and CACTUS approaches were neither designed nor adapted for tabular data in their publicly available implementations. We re-report these results from the STUNT paper,  whose authors performed such adaptation on their own. TabPFN cannot be evaluated on TEX because its public implementation supports ≤10 classes, while TEX has 11. SEBA remains the generally best-performing approach  across the expanded benchmark.

---

> ### Author Response · Authors · 2025-11-22
> **Response to Reviewer iSr3 [2/2]**
>
> ### W4: Theoretical support for nearest-neighbor alignment in a randomly selected target view
> To support empirical analysis of nearest neighbor alignment with a target task (Fig 3a & 6), we conducted a theoretical analysis taking into account classes following Gaussian distributions, see the newly added Appendix C for details. We derive that the expected value of the probability of the mismatch event (a sample and its nearest neighbor calculated in a target view come from different classes) decays exponentially as the distance between Gaussian classes increases in the original space.  Moreover, the probability of this event gets lower as the target views contain more of the original features. That is why we use masking ratios in the range [0.1, 0.5], which means that we discard at most half of the original features to compute nearest neighbors. Concluding, if the classes are separated well enough, then pretext tasks generated by SeBA are informative for a target task.
>
> ### W5: Justification for the conditioned projector
> The conditioned projector is needed because the Feature View alone does not reveal which subset of features was visible, and different masks induce representations that come from different subspaces. Without conditioning, the projector must map embeddings originating from structurally different masks using identical parameters, which leads to inconsistent alignment. Conditioning supplies the missing structural information, ensuring that each masked view is projected coherently. Notably, prior tabular joint-embedding methods such as SCARF and SubTab also operate without mask conditioning, and they consistently underperform in our SS-FSL setting; this is consistent with the intuition that mask information is important for stable alignment.
>
> ### Q1: Empirical justification and sensitivity analysis of NN alignment
> Empirical evaluation, presented in Fig 3a, confirms that SeBA objective aligns with the downstream classification task -- in a great majority of cases, the nearest neighbors found in a target view share the same class as the original instance.
> To analyze sensitivity, to k, we provide a variant of Analysis presented in Fig. 3a where the label agreement is not defined in terms of a single, but k nearest neighbors. The newly added Fig. 3b and Fig. 7 show that the immediate nearest neighbor provides the highest alignment with the target task among top k neighbors.

---

### Author Response · Authors · 2025-12-03
**Discussion summary**

Dear PCs, SAC, and ACs,

We sincerely thank all reviewers for their constructive and insightful feedback, which has led to improvements to our work.

Reviewers acknowledged our key contributions: (i) augmentation-free SSL paradigm, (ii) robustness-oriented design, (iii) lightweight yet generalizable method, (iv) effective ablations. During the rebuttal phase, we addressed all reviewers’ remarks, in particular: (i) expanding baseline coverage, (ii) clarifying dataset and reproducibility concerns, and (iii) demonstrating SeBA’s stability and competitiveness in the tabular SS-FSL setting (iv) providing theoretical justification for nearest neighbor alignment with a target task. Below is a brief per-reviewer summary.


**Reviewer iSr3.**

We expanded baseline coverage as requested by adding T-JEPA, Pseudo-Labeling, and Mean Teacher, and reported performance variance for strong baselines. We clarified the conceptual distinction between SeBA and dependency-prediction approaches and showed that SeBA remains competitive and stable in the SS-FSL setting. We provided theoretical justification and expanded experimental support for nearest neighbor alignment with a target task.

**Reviewer nyXb.**

We clarified the motivation for lightweight MLP encoders in few-shot tabular scenarios and added comparisons to attention-based models. We explained the behavior on atypical datasets such as CMC and GES and confirmed that SeBA remains considerably more computationally efficient than recent state-of-the-art baselines.

**Reviewer Jx13.**

We elaborated on the novelty of SeBA as an augmentation-free alignment method specific to tabular SSL, and expanded comparisons to transformer-based and self-supervised approaches. We also provided additional analysis of conditions in which nearest-neighbor alignment is reliable.

**Reviewer LvJK.**

We addressed baseline coverage concerns by adding SAINT under the same SS-FSL protocol, where SeBA outperforms it on all but one dataset without tuning or heavy architectures. We clarified reproducibility limitations for UMTRA/SES/CACTUs, ICT, and FESTA, and demonstrated that SeBA performs competitively as a single-encoder model, with ensembling serving only to reduce variance in extreme few-shot regimes.

The discussion period enabled us to address all major reviewer concerns through expanded baselines, additional experiments, and clarifications of scope, confirming SeBA’s suitability for tabular Semi-supervised Few-shot learning.

---

### Meta-Review · Area_Chair_oqKW · 2026-01-08

**Summary:**

This paper proposes SeBA, a semi-supervised few-shot learning framework for tabular data that generates two views by randomly splitting features and aligns their representations using consistency regularization. The method utilizes a dual-encoder architecture to maximize the agreement between these non-overlapping feature views to leverage unlabeled data.
Initial concerns focused on the method's technical novelty, the sufficiency of baseline comparisons, and the validity of the feature alignment assumption:
1. Limited Technical Novelty: Reviewers iSr3 and Jx13 pointed out that the proposed method is incremental, as it resembles existing joint-embedding architectures (like T-JEPA) or classical co-training strategies without significant algorithmic innovation.
2. Sufficiency of Baselines: Reviewers iSr3 and LvJK noted the absence of critical baselines such as SAINT, TabPFN, and standard semi-supervised methods (Mean Teacher, Pseudo-Labeling), which limit the fairness of the performance comparison.
3 . Validity of Alignment Assumption: Reviewers iSr3, nyXb, and LvJK questioned the theoretical and empirical validity of the "random splitting" strategy, arguing that nearest neighbors in partial views may not reliably reflect semantic similarity, especially in datasets with low feature redundancy.
4. Dependence on Ensembling: Reviewers LvJK and nyXb raised concerns that the method relies heavily on an ensemble of multiple separation ratios for stability, which increases computational cost and suggests that the core single-model performance is unstable.

The authors provided a rebuttal that expanded baseline comparisons (adding SAINT, T-JEPA) and offered a Gaussian-based theoretical justification. However, the recommendation tends towards rejection as significant concerns regarding the method's robustness and novelty persist. Specifically, Reviewers iSr3, Jx13, and LvJK maintained their scores at "Marginally Below Acceptance" (4), highlighting that the method's reliance on ensembling to mitigate single-model instability incurs high computational costs, and the distinction from T-JEPA remains incremental. Furthermore, the poor performance on datasets with low feature redundancy (e.g., GES) suggests the core "nearest-neighbor alignment" assumption lacks generalizability.

**Reviewer Concerns:**

1. The authors argued that the novelty lies in the "augmentation-free" paradigm and highlighted differences from T-JEPA (e.g., objective function, encoder type). This concern is Outstanding, as the core mechanism remains a standard consistency regularization setup, and the distinction from existing methods is largely implementation-based rather than a fundamental algorithmic breakthrough.
2. The authors added comparisons with T-JEPA, Pseudo-Labeling, Mean Teacher, and later included results for SAINT during the discussion period. This concern is resolved, as the authors made significant efforts to include the requested baselines and demonstrated competitive performance.
3. The authors provided a theoretical analysis (Appendix C) assuming Gaussian distributions to justify the nearest-neighbor alignment. This concern is Partially Resolved, but the reliance on strong distributional assumptions (Gaussian) does not fully alleviate the concern that random splitting is a heuristic that may fail on complex, non-redundant tabular data.
4. The authors argued that the ensemble is computationally efficient due to the use of lightweight MLPs and provides robustness. This concern is Outstanding, as Reviewer LvJK pointed out that the single-ratio performance is significantly lower and unstable, indicating that the method's effectiveness stems more from brute-force ensembling than from the quality of the learned representation itself.

**Reviewer Scores:**

- Reviewer iSr3: 4 -> 4. While concern 2 was addressed, the fundamental concern about novelty (C1) and the rationale for random masking (concern 3) likely remain.
- Reviewer nyXb: 6 -> 6. The reviewer acknowledged the rebuttal but explicitly stated that the feedback "didn't seem to have had a significant impact on my score," leaving the concern about theoretical depth (concern 3) outstanding.
- Reviewer Jx13: 4 -> 4. The concern regarding the "incremental" nature of the work (concern 1) compared to the general SSL paradigm was not fundamentally changed by the rebuttal.
- Reviewer LvJK: 4 -> 4. Although concern 2 (SAINT) was resolved, the reviewer maintained concerns about the method's instability without ensembling (concern 4), implying the core contribution is weak.

---

### Decision · Program_Chairs · 2026-01-26

Reject